# DeepWave: A Recurrent Neural-Network for Real-Time Acoustic Imaging

**Matthieu Simeoni** [*]
IBM Zurich Research Laboratory
meo@zurich.ibm.com

**Sepand Kashani** [†]
École Polytechnique Fédérale de Lausanne (EPFL)
sepand.kashani@epfl.ch

**Paul Hurley**
Western Sydney University
paul.hurley@westernsydney.edu.au

**Martin Vetterli**
École Polytechnique Fédérale de Lausanne (EPFL)
martin.vetterli@epfl.ch

## Abstract

We propose a recurrent neural-network for real-time reconstruction of acoustic camera spherical maps. The network, dubbed DeepWave, is both physically and algorithmically motivated: its recurrent architecture mimics iterative solvers from convex optimisation, and its parsimonious parametrisation is based on the natural structure of acoustic imaging problems. Each network layer applies successive filtering, biasing and activation steps to its input, which can be interpreted as generalised deblurring and sparsification steps. To comply with the irregular geometry of spherical maps, filtering operations are implemented efficiently by means of graph signal processing techniques. Unlike commonly-used imaging network architectures, DeepWave is moreover capable of directly processing the complex-valued raw microphone correlations, learning how to optimally back-project these into a spherical map. We propose moreover a smart physically-inspired initialisation scheme that attains much faster training and higher performance than random initialisation. Our real-data experiments show DeepWave has similar computational speed to the state-of-the-art delay-and-sum imager with vastly superior resolution. While developed primarily for acoustic cameras, DeepWave could easily be adapted to neighbouring signal processing fields, such as radio astronomy, radar and sonar.

## 1 Introduction

**Motivation** An *acoustic camera (AC)* [26, 8, 18, 24] is a multi-modal imaging device that allows one to visualise in real-time sound emissions from every *direction* in space. This is typically achieved by overlaying on the live video from an optical camera a heatmap representing the intensity of the ambient directional sound field, recovered from the simultaneous recordings of a *microphone array* [3, 42]. Most commercial acoustic cameras recover the sound intensity field by combining linearly the correlated microphone recordings with a *Delay-And-Sum (DAS) beamformer* [42, Chapter 5]. The beamformer acts as an *angular filter* [20, 21], steering sequentially the array sensitivity pattern –or *beamshape*– towards various directions where the sound intensity field is probed. Acoustic images obtained this way are cheap to compute, but are *blurred* by the beamshape of the microphone array,

---

[*]Corresponding author. Matthieu Simeoni is also affiliated to the École Polytechnique Fédérale de Lausanne (EPFL), with email address matthieu.simeoni@epfl.ch

[†]Matthieu Simeoni and Sepand Kashani have contributed equally to this work. Sepand Kashani was in part supported by the Swiss National Science Foundation grant number 200021 181978/1, "SESAM - Sensing and Sampling: Theory and Algorithms".

and hence exhibit poor angular resolution [49, 7, 48]. The severity of this blur can be shown [53] to be proportional to the ratio $\lambda/D$, where $D$ is the diameter of the microphone array and $\lambda$ the sound wavelength. Because of the relatively large wavelengths of acoustic waves in the audible range, this blur can be significant in practice: a 30 cm diameter microphone array has an angular resolution at 5 kHz (an $E\flat$) of approximately 10 degrees, against $7{\cdot}10^{-4}$ degrees for a standard optical camera at 790 THz (violet). Moreover, acoustic cameras are often deployed in confined environments [34], requiring them to be as *compact* and *portable* as possible, which limits[3] further the achievable angular resolution.

The advent of *compressed sensing* techniques [14, 44] –and their wide adoption in imaging sciences [54, 4, 32]– have inspired algorithmic solutions [48, 7, 11, 12] to the acoustic imaging problem, promising vastly improved angular resolutions. Unfortunately, these methods proved ill-suited for real-time purposes. Indeed, they often rely on iterative solvers, such as *proximal gradient descent (PGD)* [37] or its accelerated variants [2, 31]. While exhibiting a fast convergence rate [2], such methods still require on the order of a few dozen iterations to converge in practice, making them unable to cope with the high refresh-rate[4] of acoustic cameras. For this reason, and despite their clear superiority in terms of resolving power, nonlinear imaging methods have not yet replaced the suboptimal DAS imager in the software stack of commercial acoustic cameras.

The recent eruption of *deep learning* [33, 56, 10] in the field of imaging sciences may however seal the fate of DAS for good. Indeed, this new imaging paradigm leverages *neural-networks* [28] to reduce dramatically the image formation time. Unlike compressed-sensing methods which proceed iteratively, neural networks *encode* the image reconstruction process in a cascade of linear and nonlinear transformations *trained* on a very large number of input/output example pairs. Once properly trained, a neural-network can be efficiently evaluated for some input data to produce images of high quality, with similar accuracy and resolution as state-of-the-art compressed-sensing methods [33]. Network architectures used for inverse imaging [22, 17, 56, 10, 40] are most often *convolutional neural-networks (CNNs)*, directly adapted from generic architectures developed for *image classification* and *segmentation* [45]. While suitable for image processing tasks such as *denoising*, *super-resolution* or *deblurring* [38, 6], such architectures are ill-suited [33] for more complex image reconstruction problems where the input data may not consist of an image, as is the case in *biomedical imagery* [4, 32], *interferometry* [54] or acoustic imaging. Moreover, and particularly limiting for our current purposes, standard convolutional architectures cannot handle images with non-Euclidean domains [13] such as *spherical* maps [41] produced by omnidirectional acoustic or optical cameras.

To overcome these limitations, *recurrent* architectures [16, 50, 30, 33] have been proposed, by *unrolling* iterative convex optimisation algorithms. Such networks are not only able to handle non-image inputs, but also have greater interpretability than generic CNNs. For example, Gregor and LeCun proposed in their pioneering work [16] a *recurrent neural-network (RNN)* dubbed *LISTA*[5], inspired from the popular *iterative soft-thresholding algorithm (ISTA)*[2].[6] Their network can be seen as generalising ISTA, allowing for the normally fixed gradient and proximal steps occurring at each iteration of the algorithm to be learnt from the data: update steps of ISTA are replaced by a cascade of recurrent layers with trainable parameters. The depth of the resulting RNN is typically much smaller than the number of iterations required for ISTA to converge. Roughly speaking, the network is learning *shortcuts* in the reconstruction space, allowing it to achieve a prescribed reconstruction accuracy faster than gradient-based iterative methods.[7]

While the effectiveness of LISTA was verified on small images from the MNIST dataset (784 pixels) [16], its application to large-scale imaging problems remains challenging. This is mainly due to the huge number of weights parametrising the network which, in the fully-connected case, grows as the number of pixels to the square. Storing[8] –let alone learning– all those weights quickly becomes intractable for increasing resolutions. As a potential fix, Gregor and LeCun recommended sparsifying the network by *pruning* layer connections. While they showed that such a pruning could reduce the number of parameters in the network by 80% without affecting too much the performance of the

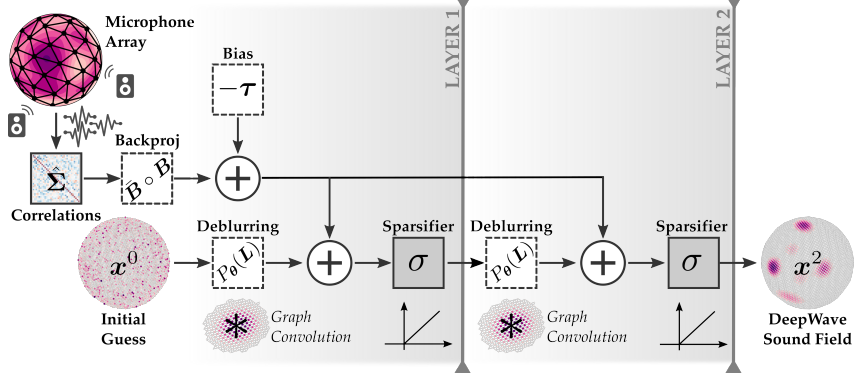

Figure 1: DeepWave's recurrent architecture (1) for $L = 2$ layers and random initialisation. Learnable parameters of the network are denoted by dashed boxes. Affine operations are denoted by white boxes and nonlinear activations by grey boxes.

latter, this is still insufficient for large-scale problems, and additional structure must be considered on network layers. Such structure is however often very dependent on the problem at hand.

**Contributions**  In this work, we propose the first realistic architecture of a LISTA neural-network adapted to acoustic imaging. Our custom architecture, dubbed *DeepWave*, is capable of rendering high-resolution spherical maps of real-life sound intensity fields in milliseconds. DeepWave is tailored to the acoustic imaging problem, leveraging fully its underlying structure so as to minimise the number of network parameters. The latter is easy to train, with a typical training time of less than an hour on a general-purpose CPU. Unlike most state-of-the-art neural-network architectures, it moreover readily supports *complex*-valued input vectors, making it capable of directly processing the raw correlated microphone recordings. Assuming a microphone array with $M$ microphones, the instantaneous covariance matrix $\hat{\mathbf{\Sigma}} \in \mathbb{C}^{M \times M}$ of the microphone recordings is processed by the network as follows (see also fig. 1):

$$\mathbf{x}^l = \sigma \left( P_{\boldsymbol{\theta}} \left( \mathbf{L} \right) \mathbf{x}^{l-1} + \left[ \overline{\mathbf{B}} \circ \mathbf{B} \right]^H \text{vec}(\hat{\mathbf{\Sigma}}) - \boldsymbol{\tau} \right), \qquad l = 1, \ldots, L, \tag{1}$$

where $\text{vec} : \mathbb{C}^{M \times M} \to \mathbb{C}^{M^2}$ is the *vectorisation operator* and $\circ$ denotes the *Khatri-Rao product* (see appendix A[9] for definitions). The neurons $\{\mathbf{x}^1, \ldots, \mathbf{x}^L\} \subset \mathbb{R}^N_+$ at the output of each layer $l$ of the depth $L$ neural-network correspond to the acoustic image as it is processed by the network, with $N$ the number of pixels. The neuron $\mathbf{x}^0 \in \mathbb{R}^N_+$ defines the initial state of the network. The nonlinear *activation function*[10] $\sigma : \mathbb{R} \to \mathbb{R}$ induces *sparsity* in the acoustic image, and is inspired by the proximal operator of an elastic-net penalty [37]. The remaining quantities, namely $P_{\boldsymbol{\theta}}(\mathbf{L})$, $\mathbf{B}$ and $\boldsymbol{\tau}$ are trainable parameters of the network, with various roles:

- *Deblurring:* the matrix $P_{\boldsymbol{\theta}}(\mathbf{L}) := \sum_{k=0}^{K} \theta_k \mathbf{L}^k \in \mathbb{R}^{N \times N}$ can be interpreted as a *deblurring matrix*, cleaning potential artefacts from the array beamshape. Following the approach of [41], it is defined as a polynomial of the *graph Laplacian* $\mathbf{L} \in \mathbb{R}^{N \times N}$ based on the *connectivity graph* of the spherical tessellation in use, with learnable coefficients $\boldsymbol{\theta} = [\theta_0, \ldots, \theta_K] \in \mathbb{R}^{K+1}$. Such parametrisation permits notably the interpretation of $P_{\boldsymbol{\theta}}(\mathbf{L})$ as a finite-support filter defined on the tessellation graph. Moreover, fast graph convolution algorithms are available for such filters [13].

- *Back-projection:* the operation $\left[ \overline{\mathbf{B}} \circ \mathbf{B} \right]^H \text{vec}(\hat{\mathbf{\Sigma}}) = \text{diag}\left( \mathbf{B}^H \hat{\mathbf{\Sigma}} \mathbf{B} \right)$ (A.8) is a *back-projection*, mapping the raw microphone correlations to the image domain. Thanks to the convenient Khatri-Rao structure, this linear operation depends only on the matrix $\mathbf{B} \in \mathbb{C}^{M \times N}$.

- *Bias:* the vector $\boldsymbol{\tau} \in \mathbb{R}^N$ is a non-uniform *bias*, boosting or shrinking the neurons of the network. Since only positive neurons are activated by the nonlinearity $\sigma$, this biasing operation helps sparsify the final acoustic image.

The total number of learnable coefficients in DeepWave is *linear* in the number of pixels. The rationale behind DeepWave's architecture is detailed in section 2, with theoretical justifications for the structures of the deblurring and back-projection linear operators. In section 3, we discuss network training, including initialisation and regularisation. We moreover derive the *forward-* and *backward-propagation* recursions[11] for our custom architecture, required for forming gradient steps. Finally, we test the architecture on synthetic as well as real data acquired with the *Pyramic array* [5, 46]. DeepWave is shown to have similar resolving power as state-of-the-art compressed-sensing methods, with a computational overhead similar to the DAS imager. To our knowledge, this is the first time a nonlinear imager of the kind achieves real-time performance on a standard computing platform. While developed primarily for acoustic cameras, DeepWave can easily be applied in neighbouring array signal processing fields [27], including radio astronomy, radar and sonar technologies.

## 2 Network architecture

In this section, we proceed similarly to [16, 50, 30] and construct DeepWave by studying the update equations of an iterative solver, namely proximal gradient descent applied to acoustic imaging.

### 2.1 Proximal gradient descent for acoustic imaging

In all that follows, we model the sound intensity field as a discrete *spherical map* with resolution $N$, specified by an *intensity vector* $\mathbf{x} \in \mathbb{R}_+^N$ and a *tessellation* $\Theta = \{\mathbf{r}_1, \ldots, \mathbf{r}_N\} \subset \mathbb{S}^2$. Spherical tessellations [19, 15] can be viewed as pixelation schemes for spherical geometries (see appendix B.1). As is customary in compressed-sensing, we propose to recover the sound intensity map by solving a convex optimisation problem (see appendix C):

$$\hat{\mathbf{x}} = \arg \min_{\mathbf{x} \in \mathbb{R}_+^N} \frac{1}{2} \left\| \hat{\boldsymbol{\Sigma}} - \mathbf{A} \operatorname{diag}(\mathbf{x}) \mathbf{A}^H \right\|_F^2 \quad + \quad \lambda \left[ \gamma \|\mathbf{x}\|_1 + (1-\gamma) \|\mathbf{x}\|_2^2 \right], \quad (2)$$

where $\|\cdot\|_F$ denotes the *Frobenius norm*, $\gamma \in ]0, 1[$ and $\lambda > 0$ are hyperparameters, and $\hat{\boldsymbol{\Sigma}} \in \mathbb{C}^{M \times M}$ is the empirical covariance matrix of the microphone recordings. In a far-field context, the *forward map* $\mathbf{A} \in \mathbb{C}^{M \times N}$ –linking the intensity vector to the microphone recordings– is commonly modelled by the so-called *steering matrix* [27]: $[\mathbf{A}]_{mn} := \exp\left(-2\pi j \langle \mathbf{p}_m, \mathbf{r}_n \rangle / \lambda_0\right)$, where $\{\mathbf{p}_1, \ldots, \mathbf{p}_M\} \subset \mathbb{R}^3$ are the microphone locations and $\lambda_0 > 0$ the sound wavelength. Using properties (A.5) and (A.6) of the *vectorisation operator* and the Frobenius norm [23, 53], problem (2) can be re-written in vectorised form as:

$$\hat{\mathbf{x}} = \arg \min_{\mathbf{x} \in \mathbb{R}_+^N} \frac{1}{2} \left\| \operatorname{vec}\left(\hat{\boldsymbol{\Sigma}}\right) - \left(\overline{\mathbf{A}} \circ \mathbf{A}\right) \mathbf{x} \right\|_2^2 \quad + \quad \lambda \left[ \gamma \|\mathbf{x}\|_1 + (1-\gamma) \|\mathbf{x}\|_2^2 \right], \quad (3)$$

where $\circ$ denotes the *Khatri-Rao product* (see definition A.3). Problem (3) is an *elastic-net penalised least-squares problem* [57], which seeks an optimal[12] trade-off between *data-fidelity* and *group-sparsity*. Group-sparsity is in this context better suited than traditional sparsity since acoustic sources are often diffuse. It is worth noting that, since the elastic-net functional is strictly convex for $\gamma \in [0, 1[$, problem (3) admits a unique solution. The latter can moreover be approximated by means of *proximal gradient descent (PGD)* [2], whose update equations are given here by (see appendix D):

$$\mathbf{x}^k = \operatorname{ReLu}\left( \frac{\mathbf{x}^{k-1} - \alpha \left(\overline{\mathbf{A}} \circ \mathbf{A}\right)^H \left[ \left(\overline{\mathbf{A}} \circ \mathbf{A}\right) \mathbf{x}^{k-1} - \operatorname{vec}\left(\hat{\boldsymbol{\Sigma}}\right) \right] - \lambda \alpha \gamma}{2 \lambda \alpha (1-\gamma) + 1} \right), \quad k \geq 1, \quad (4)$$

where $\mathbf{x}^0 \in \mathbb{R}^N$ is arbitrary, $\alpha \leq 1/\left\|\overline{\mathbf{A}} \circ \mathbf{A}\right\|_2^2$ is the *step size* and $\operatorname{ReLu}(x) := \max(x, 0)$ is the *rectified linear unit* [29], applied element-wise to a real vector.[13] The sequence of iterates $\{\mathbf{x}^k\}_{k \in \mathbb{N}}$

defined in (4) reduces the objective function in (3) at a rate $O(1/k)$ [2]. Accelerated variants of proximal gradient descent have been proposed [2], which modify (4) with an extra *momentum term*:

$$\begin{cases} \mathbf{y}^k & = \mathrm{ReLu}\left( \dfrac{\mathbf{x}^{k-1} - \alpha\left(\overline{\mathbf{A}} \circ \mathbf{A}\right)^H \left[\left(\overline{\mathbf{A}} \circ \mathbf{A}\right) \mathbf{x}^{k-1} - \mathrm{vec}\left(\hat{\boldsymbol{\Sigma}}\right)\right] - \lambda\alpha\gamma}{2\lambda\alpha(1-\gamma)+1} \right), \quad k \geq 1, \quad (5) \\ \mathbf{x}^k & = \mathbf{y}^k + \omega^k \left(\mathbf{y}^k - \mathbf{y}^{k-1}\right) \end{cases}$$

where the *momentum sequence* $\{\omega^k\}_{k\in\mathbb{N}}$ can be designed in various ways [31, 9]. In our experiments, we will use (5) as a baseline for speed comparisons, where $\omega^k$ is updated according to Chambolle and Dossal's strategy [9]: $\omega^k = (k-1)/(k+d)$, $k \geq 0$, with $d = 50$ [31]. The *accelerated proximal gradient descent (APGD)* method thus obtained is the fastest reported in the literature, with convergence rate $o(1/k^2)$ [31]. Finally, we leverage the formulae $\left(\overline{\mathbf{A}} \circ \mathbf{A}\right)\mathbf{x} = \mathrm{vec}(\mathbf{A}\,\mathrm{diag}(\mathbf{x})\mathbf{A}^H)$ (A.5), and $\left(\overline{\mathbf{A}} \circ \mathbf{A}\right)^H \mathrm{vec}(\mathbf{R}) = \mathrm{diag}(\mathbf{A}^H\mathbf{R}\mathbf{A})$ (A.8), to compute gradient steps efficiently in (5).

## 2.2 DeepWave : a PGD-inspired RNN for fast acoustic imaging

In practice PGD is terminated according to some stopping criterion. The intensity map $\mathbf{x}^L$ obtained after $L$ iterations of (4) can then be seen as the output of an RNN with depth $L$ and intermediate neurons linked by the recursion formula:

$$\mathbf{x}^l = \mathrm{ReLu}\left(\boldsymbol{\mathcal{D}}\mathbf{x}^{l-1} + \boldsymbol{\mathcal{B}}\,\mathrm{vec}\left(\hat{\boldsymbol{\Sigma}}\right) - \boldsymbol{\tau}\right), \qquad l = 1, \ldots, L. \tag{6}$$

We call this RNN the *oracle RNN*, since its weights $\boldsymbol{\mathcal{D}} \in \mathbb{R}^{N \times N}$, $\boldsymbol{\mathcal{B}} \in \mathbb{C}^{N \times M^2}$ and $\boldsymbol{\tau} \in \mathbb{R}^N$ are not learnt but simply *given* to us by identifying (6) with (4):

$$\boldsymbol{\mathcal{D}} = \frac{1}{\beta}\left[I - \alpha\left(\overline{\mathbf{A}} \circ \mathbf{A}\right)^H \left(\overline{\mathbf{A}} \circ \mathbf{A}\right)\right], \quad \boldsymbol{\mathcal{B}} = \frac{\alpha}{\beta}\left(\overline{\mathbf{A}} \circ \mathbf{A}\right)^H, \quad \boldsymbol{\tau} = \frac{\lambda\alpha\gamma}{\beta}\mathbf{1}_N, \tag{7}$$

where $\beta = 2\lambda\alpha(1-\gamma) + 1$. An analysis of (7) allows us moreover to interpret physically the affine operations performed by the oracle RNN. The matrix $\boldsymbol{\mathcal{B}}$ first is a *back-projection* operator, mapping the vectorised correlation matrix into a spherical map by applying the adjoint of the forward operator used in (3). The resulting spherical map is called a *dirty map*, and is equivalent to the DAS image [53, Section 5.2][55]. The matrix $\boldsymbol{\mathcal{D}}$ then is a *deblurring operator*, which subtracts at each iteration a fraction of the array beamshape from the spherical map, hence *cleaning* the latter of blur artefacts. The vector $\boldsymbol{\tau}$ finally is an affine *shrinkage operator*, which biases uniformly the spherical map. The latter permits –in conjunction with the rectified linear unit– the *sparsification* of the spherical map and hence improve its angular resolution.

Since the oracle RNN is merely a reinterpretation of PGD, it inherits all its properties. In particular, it is capable of solving (3) with high accuracy for arbitrary input correlation matrices. Unfortunately, this great generalisability is typically obtained at the price of a very large number[14] of layers $L$, resulting in impractical reconstruction times. If one is however willing to sacrifice some of this generalisability, it is possible to reduce drastically the network depth by unfreezing the weights $\boldsymbol{\mathcal{D}}$, $\boldsymbol{\mathcal{B}}$, $\boldsymbol{\tau}$ in (6), and allowing them to be *learnt* for some specific input distribution. This idea was first explored in the context of sparse coding by Gregor and LeCun [16], resulting in the LISTA network. A fully-connected architecture, corresponding to unconstrained $\boldsymbol{\mathcal{D}}$, $\boldsymbol{\mathcal{B}}$ and $\boldsymbol{\tau}$, would however result in $\mathcal{O}(N^2)$ weights to be learnt, which is unfeasible in large-scale acoustic imaging problems. To overcome this issue, we propose in the next paragraphs a parsimonious parametrisation of $\boldsymbol{\mathcal{D}}$ and $\boldsymbol{\mathcal{B}}$. The resulting RNN architecture, dubbed DeepWave, is given in (1) and depicted in fig. 1.

**Parametrisation of $\boldsymbol{\mathcal{D}}$** Our parametrisation of $\boldsymbol{\mathcal{D}}$ is motivated by the following result, characterising the oracle deblurring kernel for *spherical microphone arrays*[42] (see proof in appendix E).

**Proposition 1.** *Consider a* spherical microphone array, *with diameter $D$ and microphone directions* $\{\tilde{\mathbf{p}}_1, \ldots, \tilde{\mathbf{p}}_M\} \subset \mathbb{S}^2$, *forming a* near-regular tessellation *of the sphere. Then, we have*

$$\left[I - \alpha\left(\overline{\mathbf{A}} \circ \mathbf{A}\right)^H \left(\overline{\mathbf{A}} \circ \mathbf{A}\right)\right]_{ij} \simeq \left[\delta_{ij} - \alpha M^2 \mathrm{sinc}^2\left(\frac{D}{\lambda_0}\|\mathbf{r}_i - \mathbf{r}_j\|\right)\right], \forall i, j \in \{1, \ldots, N\} \tag{8}$$

| **Algorithm 1** DeepWave forward propagation | **Algorithm 2** DeepWave backward propagation |
|---|---|
| 1: **Input**: $\hat{\Sigma}_t$, $\mathbf{x}_t^0$, $\hat{\mathbf{x}}_t$, $\boldsymbol{\theta}$, $\mathbf{B}$, $\boldsymbol{\tau}$, $\sigma$ | 1: **Input**: $\hat{\Sigma}_t$, $\mathbf{x}_t^0$, $\hat{\mathbf{x}}_t$, $\boldsymbol{\theta}$, $\mathbf{B}$, $\sigma$, $\left\{\mathbf{s}_t^l\right\}_{l=1,\ldots,L}$ |
| 2: **Output**: $\mathcal{L}_t \in \mathbb{R}_+$, $\left\{\mathbf{s}_t^l\right\}_{l=1,\ldots,L} \subset \mathbb{R}^N$ | 2: **Output**: $\partial\boldsymbol{\theta} \in \mathbb{R}^{K+1}$, $\partial\mathbf{B} \in \mathbb{C}^{M\times N}$, $\partial\boldsymbol{\tau} \in \mathbb{R}^N$ |
| 3: | 3: $(\partial\mathbf{x}, \partial\boldsymbol{\theta}, \partial\boldsymbol{\tau}) \leftarrow \left(\left(\sigma(\mathbf{s}_t^L) - \hat{\mathbf{x}}_t\right)/\|\hat{\mathbf{x}}_t\|_2^2, \mathbf{0}, \mathbf{0}\right)$ |
| 4: $\mathbf{y}_t \leftarrow \mathrm{diag}(\mathbf{B}^H\hat{\Sigma}_t\mathbf{B}) - \boldsymbol{\tau}$ | 4: **for** $l$ **in** $[L,\ldots,1]$ **do** |
| 5: **for** $l$ **in** $[1,\ldots,L]$ **do** | 5: $\quad \partial\mathbf{s} \leftarrow \mathrm{diag}\left(\sigma'(\mathbf{s}_t^l)\right)\partial\mathbf{x}$ |
| 6: $\quad \mathbf{s}_t^l \leftarrow P_{\boldsymbol{\theta}}(\mathbf{L})\mathbf{x}_t^{l-1} + \mathbf{y}_t$ | 6: $\quad \partial\mathbf{x} \leftarrow P_{\boldsymbol{\theta}}(\mathbf{L})\partial\mathbf{s}$ |
| 7: $\quad \mathbf{x}_t^l \leftarrow \sigma(\mathbf{s}_t^l)$ | 7: $\quad \partial\boldsymbol{\tau} \leftarrow \partial\boldsymbol{\tau} - \partial\mathbf{s}$ |
| 8: $\mathcal{L}_t \leftarrow \frac{1}{2}\left\|\hat{\mathbf{x}}_t - \mathbf{x}_t^L\right\|_2^2 / \|\hat{\mathbf{x}}_t\|_2^2$ | 8: $\quad [\partial\boldsymbol{\theta}]_k \leftarrow [\partial\boldsymbol{\theta}]_k + \partial\mathbf{s}^T T_k(\mathbf{L})\,\sigma(\mathbf{s}_t^{l-1})$ |
|  | 9: $\partial\mathbf{B} \leftarrow -2\hat{\Sigma}_t\mathbf{B}\,\mathrm{diag}\left(\partial\boldsymbol{\tau}\right)$ |

Figure 2: Forward and backward algorithms to compute gradients of $\mathcal{L}_t$ with respect to $\boldsymbol{\theta}, \mathbf{B}, \boldsymbol{\tau}$. For notational simplicity we use the shorthand $\partial\boldsymbol{\alpha} = \partial\mathcal{L}_t/\partial\boldsymbol{\alpha}$, and assume $\sigma(\mathbf{s}_t^0) = \mathbf{x}_t^0$.

*where $\lambda_0$ is the* wavelength, *$\delta_{ij}$ denotes the* Kronecker delta *and* $\mathrm{sinc}(x) := \sin(\pi x)/\pi x$ *is the* cardinal sine. *Moreover, the approximation* (8) *is* extremely good *for* $M \geq 3\lfloor\frac{2\pi D}{\lambda_0}\rfloor^2$.

Proposition 1 tells us that, for spherical arrays with sufficient number of microphones[15], the oracle deblurring operator $\mathcal{D}$ in (7) corresponds actually to a sampled *zonal kernel* [35]: $[\mathcal{D}]_{ij} = \kappa(\|\mathbf{r}_i - \mathbf{r}_j\|)$ for some $\kappa : \mathbb{R}_+ \to \mathbb{R}$. Since zonal kernels are used to define spherical convolutions [35], $\mathcal{D}$ can hence be seen as a *discrete convolution operator* over the tessellation in use $\Theta = \{\mathbf{r}_1,\ldots,\mathbf{r}_N\}$. Its bandwidth is moreover essentially finite, since coefficients $[\mathcal{D}]_{ij}$ decay as $1/\|\mathbf{r}_i - \mathbf{r}_j\|^2$. As discussed in [41, 13], discrete spherical convolution operators with finite scope can be efficiently represented and implemented by means of *graph signal processing* [47] techniques. This leads us to consider the following parametrisation (see appendix B.3 for details): $\mathcal{D} = P_{\boldsymbol{\theta}}(\mathbf{L}) := \sum_{k=0}^K \theta_k \mathbf{L}^k$, where $\boldsymbol{\theta} = [\theta_0,\ldots,\theta_K] \in \mathbb{R}^{K+1}$, $K$ controls the scope of the discrete convolution and $\mathbf{L} \in \mathbb{R}^{N\times N}$ is the *Laplacian* [47] associated to the convex-hull graph of $\Theta$. Note that with this parametrisation, the number of parameters characterising $\mathcal{D}$ drops from $N^2$ to $K+1$, with $K \ll N$.

**Parametrisation of $\mathcal{B}$** The oracle back-projection operator (7) admits a factorisation in terms of the Khatri-Rao product. We decide hence to equip $\mathcal{B}$ with a similar structure: $\mathcal{B} = (\overline{\mathbf{B}} \circ \mathbf{B})^H$ for some learnable matrix $\mathbf{B} \in \mathbb{C}^{M\times N}$. With such a parametrisation, the number of parameters characterising $\mathcal{B}$ drops from $NM^2$ to $NM$. The Khatri-Rao structure guarantees moreover real-valued –and hence physically-interpretable– dirty maps.

## 3 Network training

To facilitate the description of the training procedure, we adopt the following shorthand notations.

- $\mathrm{DeepWave}(\boldsymbol{\Omega}, L)$ denotes a specific instance of the DeepWave network (1) with parameters $\boldsymbol{\Omega} := \{\boldsymbol{\theta}, \mathbf{B}, \boldsymbol{\tau}\}$ and depth $L$.
- $\mathrm{APGD}(\alpha, \lambda, \gamma)$ denotes an instance of APGD (5), with tuning parameters $(\alpha, \lambda, \gamma) \in \mathbb{R}_+^3$.

The network parameters are chosen as minimisers of the following optimisation problem:

$$\hat{\boldsymbol{\Omega}} \in \underset{\substack{\boldsymbol{\theta}\in\mathbb{R}^{K+1} \\ \mathbf{B}\in\mathbb{C}^{M\times N} \\ \boldsymbol{\tau}\in\mathbb{R}^N}}{\arg\min} \frac{1}{T}\sum_{t=1}^T \underbrace{\frac{\left\|\hat{\mathbf{x}}_t - \mathbf{x}_t^L(\boldsymbol{\Omega})\right\|_2^2}{2\|\hat{\mathbf{x}}_t\|_2^2}}_{:=\mathcal{L}_t} + \underbrace{\frac{\lambda_{\boldsymbol{\theta}}}{2(K+1)}\|\boldsymbol{\theta}\|_2^2}_{:=\mathcal{L}_{\boldsymbol{\theta}}} + \underbrace{\frac{\lambda_{\mathbf{B}}}{2MN}\|\mathbf{B}\|_F^2}_{:=\mathcal{L}_{\mathbf{B}}} + \underbrace{\frac{\lambda_{\boldsymbol{\tau}}}{2N}\left\|\mathbf{L}^{1/2}\boldsymbol{\tau}\right\|_2^2}_{:=\mathcal{L}_{\boldsymbol{\tau}}}.$$

(9)

The quantities $\{\mathbf{x}_t^L(\boldsymbol{\Omega})\}_t$ and $\{\hat{\mathbf{x}}_t\}_t$ in (9) correspond respectively to the outputs of $\mathrm{DeepWave}(\boldsymbol{\Omega}, L)$ and $\mathrm{APGD}(\alpha, \lambda, \gamma)$ with identical example input data $\{(\hat{\Sigma}_t, \mathbf{x}_t^0)\}_t$. The first term $\frac{1}{T}\sum_{t=1}^T \mathcal{L}_t$ is a

data-fidelity term, which attempts to bring $\hat{\mathbf{x}}_t$ and $\mathbf{x}_t^L(\mathbf{\Omega})$ as close as possible from one another.[16] The additional terms $\mathcal{L}_{\boldsymbol{\theta}}, \mathcal{L}_{\mathbf{B}}, \mathcal{L}_{\boldsymbol{\tau}}$ are smoothing *regularisers*, fighting against *overfitting*, a common issue in deep learning. Since the shrinkage operator $\boldsymbol{\tau}$ is defined over an irregular spherical tessellation, the smoothing term $\mathcal{L}_{\boldsymbol{\tau}}$ is defined via the Laplacian $\mathbf{L} \in \mathbb{R}^{N \times N}$ associated to the connectivity graph of the tessellation, as is customary in graph signal processing (see appendix B.3).

Optimisation of (9) is carried out by *stochastic gradient descent (SGD)* with momentum acceleration [51]. Gradients of $\mathcal{L}_t$ with respect to $\boldsymbol{\theta}, \mathbf{B}, \boldsymbol{\tau}$ are efficiently evaluated using *reverse-mode algorithmic differentiation* [1, 25] and are given in algorithms 1 and 2 (see appendix F for a derivation). While random initialisation of neural-networks is a common practice in deep learning [51], this strategy failed for our specific architecture, leading to poor validation loss and considerably increased training times. Instead, we hence use the oracle parameters (7) to initialise SGD:

$$\boldsymbol{\theta}^0 := \underset{\boldsymbol{\theta} \in \mathbb{R}^{K+1}}{\arg\min} \|P_{\boldsymbol{\theta}}(\mathbf{L}) - \mathcal{D}\|_F^2, \quad \mathbf{B}^0 := \sqrt{\frac{\alpha}{\beta}}\mathbf{A}, \quad \boldsymbol{\tau}^0 := \frac{\lambda\alpha\gamma}{\beta}\mathbf{1}_N. \tag{10}$$

For greater numerical stability during training, we follow [41] and reparameterise the deblurring filter as $P_{\boldsymbol{\theta}}(\tilde{\mathbf{L}}) = \sum_{k=0}^{K} \theta_k T_k(\tilde{\mathbf{L}})$, where $T_k(\cdot)$ is the Chebychev polynomial of order $k$ and $\tilde{\mathbf{L}}$ is the normalised Laplacian with spectrum in $[-1, 1]$ (see appendix B.3 for implementation details). Finally, we substitute the ReLu activation function by a scaled rectified tanh to avoid the exploding gradient problem [39].[17]

## 4 Experimental results

In this section, we compare the accuracy, resolution and runtime performance of DeepWave to DAS and APGD on real-world (RW) and simulated (SIM) datasets. More comprehensive dataset descriptions and additional results, including an ablation study, are provided in appendices G to I.

**Dataset 1 [36] (RW)**  reproduces a conference room setup depicted in figs. 3a and 3b, where 8 people[18] are gathered around a table and speak either in turns or simultaneously (with at most 3 concurrent speakers). Recordings of the conversation are collected by the 48-element Pyramic array [46] (fig. 3f) positioned at the centre of the table. Since human speech is wide-band, the audible range $[1500, 4500]$ Hz in the latter are pre-processed every 100 ms and split into 9 uniform bins to form a suitable training set $\{(\hat{\mathbf{\Sigma}}_t, \hat{\mathbf{x}}_t, \mathbf{x}_t^0)\}_t$ of 2760 data points per frequency band for DeepWave (with $N = 2234$). (See appendix G.2.) Frequency channels are processed independently by each algorithm. DeepWave is trained by splitting the data points into a training and validation set (respectively 80% and 20% in size). For each frequency band, we chose an architecture with 5 layers.

In fig. 3, figs. G.4 and H.3 respectively, we compare the accuracy and runtime of DeepWave, DAS and APGD. A video showing the evolution in time of DeepWave and DAS azimuthal sound fields (as in figs. 3a and 3b) is also available.[19] In terms of resolution, DeepWave and APGD perform similarly, outperforming DAS by approximately 27%. The mean contrast scores for DeepWave and DAS over the test set of Dataset 1 are 0.99 ($\pm0.0081$) and 0.89 ($\pm0.07$), respectively. Note that since the metrics used for assessing resolution and contrast[20] are not perfectly reflective of human-eye perception, the reported image quality improvements appear even more striking through visual inspection of the sound intensity fields (see for example fig. 3).

**Dataset 2 [43] (RW)**  consists of 2700 template recordings from the Pyramic array taken in an anechoic chambre at an angular resolution of 2 degrees in azimuth and three different elevations (-15, 0, 15 degrees). Recordings contain both male and female speech samples to cover a wide audible range. The audio samples can be combined to simulate complex multi-source sound fields, hence we leverage this property to augment the dataset to 5700 distinct recordings with one, two, or three active speakers simultaneously. The raw time-series are then pre-processed as for Dataset 1 to obtain a

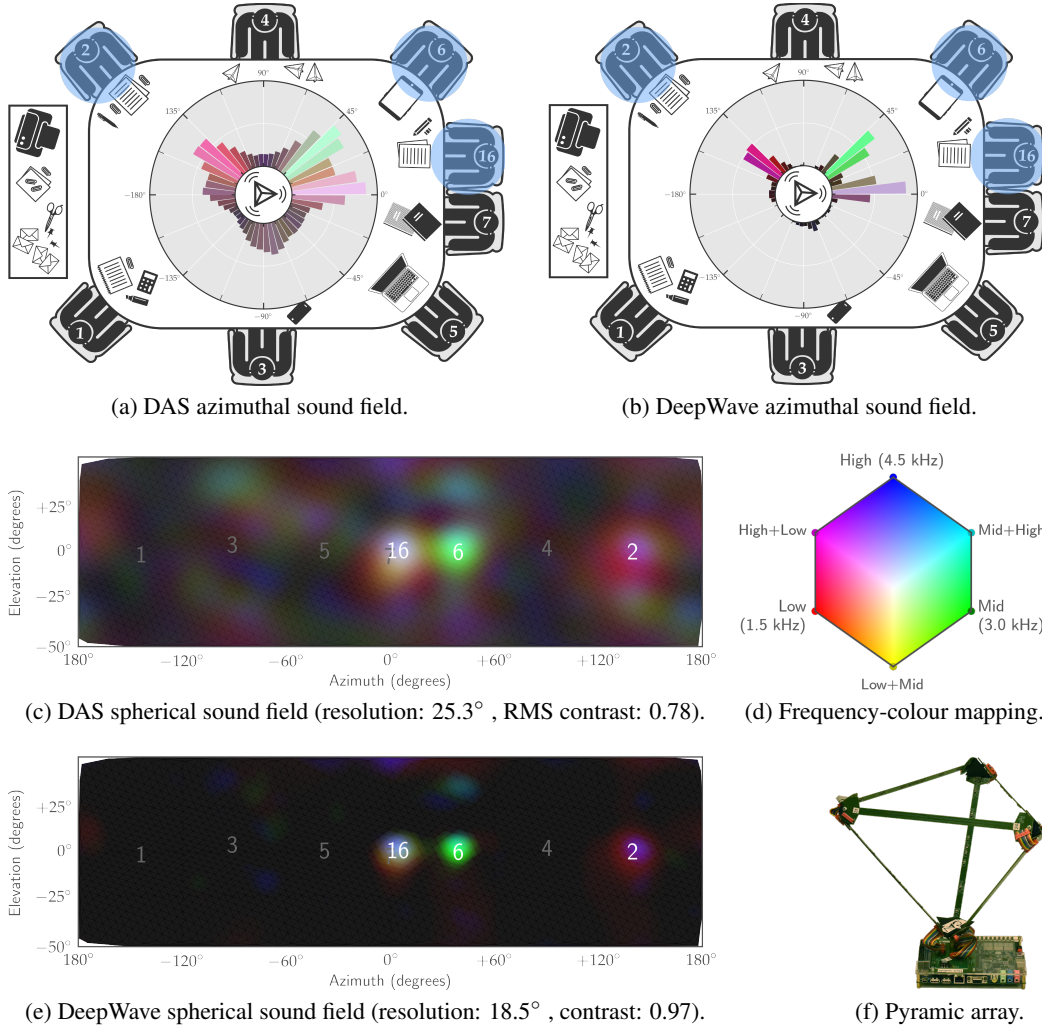

(a) DAS azimuthal sound field.

(b) DeepWave azimuthal sound field.

(c) DAS spherical sound field (resolution: 25.3° , RMS contrast: 0.78).

(d) Frequency-colour mapping.

(e) DeepWave spherical sound field (resolution: 18.5° , contrast: 0.97).

(f) Pyramic array.

Figure 3: Snapshots at time $t = 1.7$ s of the sound intensity fields produced by DeepWave and DAS for the Pyramic recordings with speakers 2, 6 and 16 active. Sound frequencies range from 1.5 to 4.5 kHz and were mapped to true colours (see fig. 3d, colour shades correspond to lower intensities). The spherical maps of DAS and DeepWave are plotted in figs. 3c and 3e, respectively. In figs. 3a and 3b we plot the azimuthal projections of figs. 3c and 3e, respectively.

training set of 151980 data points per frequency band (with $N = 1568$). Network training is identical to that of Dataset 1, except that 10 azimuth directions are also witheld from the training set to assess how well the network generalises to emissions from unseen directions.

Figures 4a and 4b show sample DAS and DeepWave reconstructions with real sources from directions withheld from the training set. Similarly, fig. 4c shows sample reconstructions when the network is trained on real data but tested on synthetic narrow-band covariance matrices induced by sources from directions absent from the training set. In both cases we see that DeepWave outperforms DAS in resolution and contrast (i.e. sharper blobs and darker background).

**Dataset 3 (SIM)** finally is a dataset with recordings from a spherical microphone array using a narrow-band point-source data-model at 2 kHz [53]. The sources are randomly positioned over a 120° field-of-view, with up to 10 concurrent sources per recording. Experiment results available in fig. H.1 corroborate the real-data results, hence showing that DeepWave generalises well to a large number of sources with unconstrained positions. We further investigated in fig. H.2 the influence of network depth, and concluded that 5 or 6 layers are generally sufficient for the investigated dataset.

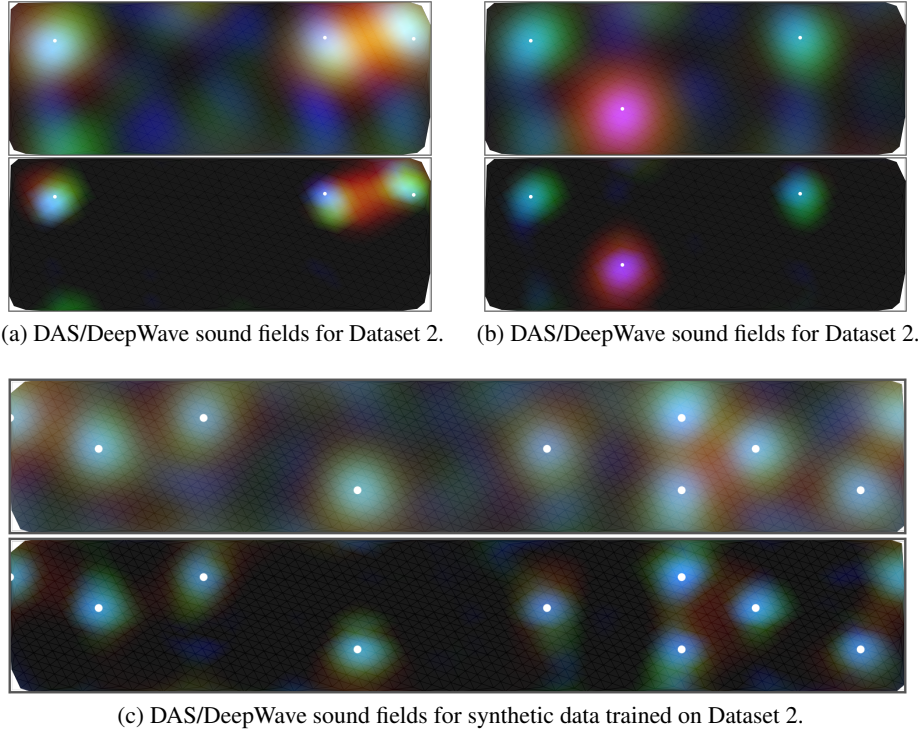

(a) DAS/DeepWave sound fields for Dataset 2.  (b) DAS/DeepWave sound fields for Dataset 2.

(c) DAS/DeepWave sound fields for synthetic data trained on Dataset 2.

Figure 4: Snapshots of the sound intensity fields produced by DeepWave and DAS when trained on Dataset 2 (with 10 held-out source directions). Each subplot contains a DAS image (top) and a DeepWave image (bottom). The frequency color mapping is identical to fig. 3d. Figures 4a and 4b show azimuthal sound field slices on $[-20°, 150°]$ using real-world covariance matrices with sources from unseen directions during training. Figure 4c shows a full $360°$ sound field on a synthetic covariance matrix from unseen directions during training. Elevations span $[-15°, +15°]$.

In terms of runtimes finally, DeepWave and DAS both reach real-time requirements (6.5 ms and 2.0 ms respectively), largely outperforming APGD (211 ms). (See fig. H.3 for more details.)

## 5 Conclusion

We introduced DeepWave, the first recurrent neural-network for real-time and high resolution acoustic imaging. It mimics iterative solvers from convex optimisation, while using the natural structure of acoustic imaging problems for efficient training and operation. Our real and simulated data experiments show DeepWave has similar computational speed to the state-of-the-art DAS imager with vastly superior resolution and contrast.

For future work, one of our goals is to make DeepWave *time-aware*, by training it on sequences of consecutive measurements in time. To this end, we plan to connect multiple DeepWave networks together, one for each time, and train them end-to-end. In such an architecture, the output neurons from one network would serve as initial neural state $\mathbf{x}^0$ for the next network in line. This can be interpreted as warm-starting the network with the sound field estimated at the previous time instant. Additionally, we would like to propose a frequency-invariant DeepWave architecture, allowing to train a single network for all frequency bands. Properties of the oracle weights (7) suggest that this should be possible. This would considerably facilitate the training of the network, since the training set would be augmented and the number of trainable parameters reduced.

**Acknowledgments** We thank Erwan Zerhouni for useful discussions regarding network training and implementation details; and Ivan Dokmanić for insights on related works that inspired our approach. Finally we express our gratitude towards Robin Scheibler and Hanjie Pan for their openly-accessible real-world datasets [36, 43].

## Footnotes

[3]Remember that the blur spread is inversely proportional to the microphone array diameter.

[4]An acoustic camera typically updates the acoustic image a dozen times per second.

[5]LISTA stands for learned iterative soft-thresholding algorithm.

[6]ISTA is an instance of proximal gradient descent for *penalised basis pursuit* problems [52].

[7]Of course, such shortcuts will most likely only be valid for the distribution of inputs and outputs implicitly defined by the training set, which should hence be carefully crafted for the network to generalise well in practice.

[8]For a 1 megapixel image, the weights parametrising the network would be approximately 8 Gb in size.

[9]In all that follows, labels prefixed with roman letters refer to elements of the supplementary material.

[10]Typified by a rectilinear unit.

[11]DeepWave implementation can be found on `https://github.com/imagingofthings/DeepWave`.

[12]The notion of optimality is defined here by the penalty parameter $\lambda$.

[13]Note that with $\mathbf{x}^0 \in \mathbb{R}^N$, every gradient step produces a real vector.

[14]Even with momentum acceleration, PGD typically requires more than 50 iterations to converge. The oracle RNN obtained by unrolling PGD will consequently be very deep.

[15]For a spherical array with diameter $D = 30$ cm operating at 1 kHz, $M \geq 90$ is sufficient.

[16]in a mean relative squared-error sense.

[17]An alternative is to use a *truncated* ReLu. Given initialisation strategy 10, network training will still converge with similar step sizes as those used with tanh non-linearities.

[18]The 8 people are represented in the experiment by loadspeakers playing male and female speech samples.

[19]Available online: https://www.youtube.com/watch?v=PwB3CS2rHdI

[20]As is customary, resolution is measured as the *width at half-maximum* of the impulse response of the algorithms. Contrast is measured as the difference between the maximum and mean of the greyscale image.

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
