[Supplementary Material · supplementary_material.pdf]

# DeepWave: A Recurrent Neural-Network for Real-Time Acoustic Imaging
## *Supplementary Material*

**Matthieu Simeoni** [*]
IBM Zurich Research Laboratory
meo@zurich.ibm.com

**Sepand Kashani** [†]
École Polytechnique Fédérale de Lausanne (EPFL)
sepand.kashani@epfl.ch

**Paul Hurley**
Western Sydney University
paul.hurley@westernsydney.edu.au

**Martin Vetterli**
École Polytechnique Fédérale de Lausanne (EPFL)
martin.vetterli@epfl.ch

## Contents

[*]Corresponding author. Matthieu Simeoni is also affiliated to the École Polytechnique Fédérale de Lausanne (EPFL), with email address matthieu.simeoni@epfl.ch

[†]Matthieu Simeoni and Sepand Kashani have contributed equally to this work. Sepand Kashani was in part supported by the Swiss National Science Foundation grant number 200021 181978/1, "SESAM - Sensing and Sampling: Theory and Algorithms".

# A   Linear algebra primer

This work makes heavy use of the Kronecker product and related operators. To ease the user's understanding, we provide a short description of these operators along with proofs of common transforms used throughout the text. Useful references for this section are [9, 5].

## A.1   Conventions

Throuhought this document, we adopt the following conventions:

- Vectors are denoted with bold lowercase letters: $\mathbf{y}$.
- Matrices are denoted with bold uppercase letters: $\mathbf{A}$.
- If $\mathbf{A} \in \mathbb{C}^{M \times N}$, $\mathbf{a}_k \in \mathbb{C}^M$ denotes the $k$-th column of $\mathbf{A}$.
- The $i$-th entry of vector $\mathbf{y}$ is denoted $[\mathbf{y}]_i$.
- The $(i, j)$-th entry of matrix $\mathbf{A}$ is denoted $[\mathbf{A}]_{ij}$.
- The conjugation operation is denoted by overlining a vector or a matrix respectively: $\overline{\mathbf{a}}, \overline{\mathbf{A}}$.
- The modulus of a complex number $z \in \mathbb{C}$ is denoted by $|z|$.

## A.2   Hadamard, Kronecker and Khatri-Rao products

The Hadamard product is the element-wise multiplication operator:

**Definition A.1** (Hadamard product). *Let* $\mathbf{A} \in \mathbb{C}^{M \times N}$ *and* $\mathbf{B} \in \mathbb{C}^{M \times N}$. *The* Hadamard product $\mathbf{A} \odot \mathbf{B} \in \mathbb{C}^{M \times N}$ *is defined as*

$$[\mathbf{A} \odot \mathbf{B}]_{ij} = [\mathbf{A}]_{ij} [\mathbf{B}]_{ij}.$$

*Moreover, we denote by* $\mathbf{A}^{\odot 2}$ *the Hadamard square of a matrix:* $\mathbf{A} \odot \mathbf{A}$.

The Kronecker product generalises the vector outer product to matrices, and represents the tensor product between two finite-dimensional linear maps:

**Definition A.2** (Kronecker product). *Let* $\mathbf{A} \in \mathbb{C}^{M_1 \times N_1}$ *and* $\mathbf{B} \in \mathbb{C}^{M_2 \times N_2}$. *The* Kronecker product $\mathbf{A} \otimes \mathbf{B} \in \mathbb{C}^{M_1 M_2 \times N_1 N_2}$ *is defined as*

$$\mathbf{A} \otimes \mathbf{B} = \begin{bmatrix} [\mathbf{A}]_{11} \mathbf{B} & \cdots & [\mathbf{A}]_{1N_1} \mathbf{B} \\ \vdots & \ddots & \vdots \\ [\mathbf{A}]_{M_1 1} \mathbf{B} & \cdots & [\mathbf{A}]_{M_1 N_1} \mathbf{B} \end{bmatrix}.$$

The main properties of the Kronecker product are [9]:

$$(\mathbf{A} \otimes \mathbf{B})^H = \mathbf{A}^H \otimes \mathbf{B}^H, \tag{A.1}$$

$$(\mathbf{A} \otimes \mathbf{B})(\mathbf{C} \otimes \mathbf{D}) = (\mathbf{AC}) \otimes (\mathbf{BD}), \tag{A.2}$$

$$(\mathbf{A} \otimes \mathbf{B}) \odot (\mathbf{C} \otimes \mathbf{D}) = (\mathbf{A} \odot \mathbf{C}) \otimes (\mathbf{B} \odot \mathbf{D}). \tag{A.3}$$

The Khatri-Rao product finally, is a column-wise Kronecker product:

**Definition A.3** (Khatri-Rao product). *Let* $\mathbf{A} \in \mathbb{C}^{M_1 \times N}$ *and* $\mathbf{B} \in \mathbb{C}^{M_2 \times N}$. *The* Khatri-Rao product $\mathbf{A} \circ \mathbf{B} \in \mathbb{C}^{M_1 M_2 \times N}$ *is defined as*

$$\mathbf{A} \circ \mathbf{B} = [\mathbf{a}_1 \otimes \mathbf{b}_1, \ldots, \mathbf{a}_N \otimes \mathbf{b}_N].$$

## A.3   Matrix identities

In imaging problems, $\mathbf{A} \otimes \mathbf{B}$ and $\mathbf{A} \circ \mathbf{B}$ are often too large to be stored in memory. However it is not the matrix itself that is of interest in many circumstances, but rather the effect of a linear map such as $f(\mathbf{x}) = (\mathbf{A} \otimes \mathbf{B})\mathbf{x}$. The matrix identities below allow us to evaluate $f(\mathbf{x})$ without ever having to compute large intermediate arrays. They make use of the vectorisation operator, defined hereafter:

**Definition A.4** (Vectorisation). *Let* $\mathbf{A} \in \mathbb{C}^{M \times N}$. *The* vectorisation *operator* $\mathrm{vec}(\cdot)$ *reshapes a matrix into a vector by stacking its columns:*

$$[\mathrm{vec}(\mathbf{A})]_{M(j-1)+i} = [\mathbf{A}]_{ij}.$$

*Conversely, the* matricisation *operator* $\mathrm{mat}_{M,N}(\cdot)$ *reshapes a vector into a matrix:*

$$[\mathrm{mat}_{M,N}(\mathbf{a})]_{ij} = [\mathbf{a}]_{M(j-1)+i}.$$

Commonly used matrix identities are the following [5, 23]:

$$\mathrm{vec}(\mathbf{ABC}) = \left(\mathbf{C}^T \otimes \mathbf{A}\right) \mathrm{vec}(\mathbf{B}) \tag{A.4}$$

$$\mathrm{vec}(\mathbf{A} \, \mathrm{diag}(\mathbf{b})\mathbf{C}) = \left(\mathbf{C}^T \circ \mathbf{A}\right) \mathbf{b} \tag{A.5}$$

$$\langle \mathbf{A}, \mathbf{B} \rangle_F = \mathrm{tr}\left(\mathbf{A}^H \mathbf{B}\right) = \mathrm{vec}(\mathbf{A})^H \, \mathrm{vec}(\mathbf{B}) \tag{A.6}$$

$$\mathrm{vec}(\mathbf{b}\mathbf{a}^T) = \mathbf{a} \otimes \mathbf{b} \tag{A.7}$$

In this work, we furthermore make use of the following nonstandard matrix identities, proven hereafter:

$$(\mathbf{A} \circ \mathbf{B})^H \, \mathrm{vec}(\mathbf{C}) = \mathrm{diag}\left(\mathbf{B}^H \mathbf{C} \overline{\mathbf{A}}\right) \tag{A.8}$$

$$(\mathbf{A} \otimes \mathbf{B})^H \, (\mathbf{A} \otimes \mathbf{B}) \, \mathrm{vec}(\mathbf{C}) = \mathrm{vec}(\mathbf{B}^H \mathbf{B} \mathbf{C} \mathbf{A}^T \overline{\mathbf{A}}) \tag{A.9}$$

$$(\mathbf{A} \circ \mathbf{B})^H \, (\mathbf{A} \circ \mathbf{B}) \, \mathbf{c} = \mathrm{diag}(\mathbf{B}^H \mathbf{B} \, \mathrm{diag}(\mathbf{c}) \mathbf{A}^T \overline{\mathbf{A}}) \tag{A.10}$$

$$(\mathbf{A} \circ \mathbf{B})^H \, (\mathbf{A} \circ \mathbf{B}) = \mathbf{A}^H \mathbf{A} \odot \mathbf{B}^H \mathbf{B}. \tag{A.11}$$

*Proof.* (A.8)

$$\left[(\mathbf{A} \circ \mathbf{B})^H \, \mathrm{vec}(\mathbf{C})\right]_i = \langle [\mathbf{A} \circ \mathbf{B}]_i \,, \mathrm{vec}(\mathbf{C}) \rangle = (\mathbf{a}_i \otimes \mathbf{b}_i)^H \, \mathrm{vec}(\mathbf{C})$$

$$\overset{(A.7)}{=} \mathrm{vec}(\mathbf{b}_i \mathbf{a}_i^T)^H \, \mathrm{vec}(\mathbf{C}) \overset{(A.6)}{=} \mathrm{tr}\left(\overline{\mathbf{a}}_i \mathbf{b}_i^H \mathbf{C}\right)$$

$$= \mathrm{tr}\left(\mathbf{b}_i^H \mathbf{C} \overline{\mathbf{a}}_i\right) = \left[\mathbf{B}^H \mathbf{C} \overline{\mathbf{A}}\right]_{ii} = \left[\mathrm{diag}\left(\mathbf{B}^H \mathbf{C} \overline{\mathbf{A}}\right)\right]_i$$

$\square$

*Proof.* (A.9)

$$(\mathbf{A} \otimes \mathbf{B})^H \, (\mathbf{A} \otimes \mathbf{B}) \, \mathrm{vec}(\mathbf{C}) \overset{(A.1)}{=} \left(\mathbf{A}^H \otimes \mathbf{B}^H\right) (\mathbf{A} \otimes \mathbf{B}) \, \mathrm{vec}(\mathbf{C})$$

$$\overset{(A.3)}{=} \left[\left(\mathbf{A}^H \mathbf{A}\right) \otimes \left(\mathbf{B}^H \mathbf{B}\right)\right] \mathrm{vec}(\mathbf{C})$$

$$\overset{(A.4)}{=} \mathrm{vec}(\mathbf{B}^H \mathbf{B} \mathbf{C} \mathbf{A}^T \overline{\mathbf{A}})$$

$\square$

*Proof.* (A.10)

$$(\mathbf{A} \circ \mathbf{B})^H \, (\mathbf{A} \circ \mathbf{B}) \, \mathbf{c} \overset{(A.5)}{=} (\mathbf{A} \circ \mathbf{B})^H \, \mathrm{vec}\left(\mathbf{B} \, \mathrm{diag}(\mathbf{c}) \mathbf{A}^T\right)$$

$$\overset{(A.8)}{=} \mathrm{diag}\left(\mathbf{B}^H \mathbf{B} \, \mathrm{diag}(\mathbf{c}) \mathbf{A}^T \overline{\mathbf{A}}\right)$$

$\square$

*Proof.* (A.11)

$$\left[(\mathbf{A} \circ \mathbf{B})^H \, (\mathbf{A} \circ \mathbf{B})\right]_{ij} = \langle \mathbf{a}_i \otimes \mathbf{b}_i, \mathbf{a}_j \otimes \mathbf{b}_j \rangle \overset{(A.7)}{=} \langle \mathrm{vec}(\mathbf{b}_i \mathbf{a}_i^T), \mathrm{vec}(\mathbf{b}_j \mathbf{a}_j^T) \rangle$$

$$\overset{(A.6)}{=} \mathrm{tr}\left(\overline{\mathbf{a}}_i \mathbf{b}_i^H \mathbf{b}_j \mathbf{a}_j^T\right) = \mathrm{tr}\left(\mathbf{b}_i^H \mathbf{b}_j \mathbf{a}_j^T \overline{\mathbf{a}}_i\right)$$

$$= \langle \mathbf{b}_i, \mathbf{b}_j \rangle \langle \mathbf{a}_i, \mathbf{a}_j \rangle$$

When put in matrix form, the above yields

$$(\mathbf{A} \circ \mathbf{B})^H \, (\mathbf{A} \circ \mathbf{B}) = \mathbf{A}^H \mathbf{A} \odot \mathbf{B}^H \mathbf{B}$$

$\square$

(a) Equal-angle tessellation.　　(b) HEALPix tessellation.　　(c) Fibonacci tessellation.

Figure B.1: Examples of spherical maps defined over tessellations on the sphere, with an approximate resolution of $N = 200$ for each scheme. Cell centres are marked by black dots. The equal-angle tessellation fig. B.1a is obtained by pixelating the azimuth-elevation domain. The Fibonacci tessellation fig. B.1c is obtained by constructing the spherical Voronoi tessellation of the Fibonacci lattice (B.1). The HEALPix tessellation fig. B.1b finally, very popular in cosmology and astronomy, is constructed by hierarchical subdivision of the Voronoi cells of the dodecahedron vertices [4].

# B  Signal processing tools for spherical maps

## B.1  Spherical tessellations and spherical maps

In the Euclidean setting, images are commonly stored, manipulated and displayed as multi-dimensional arrays. This convenient data-structure implicitly assumes a uniform partitioning of the image support into rectangular tiles or *pixels*. Spherical maps on the other hand are stored as arbitrarily ordered intensity vectors $\mathbf{x} \in \mathbb{R}^N$ associated to a *lattice* of directions $\Theta = \{\mathbf{r}_1, \ldots, \mathbf{r}_N\} \subset \mathbb{S}^2$ sampling irregularly the sphere. The pixels are obtained as polygonal Voronoi cells of the set $\Theta$, and form a *spherical tessellation* [7]. For example, the equal-angle tessellation in fig. B.1a is obtained by pixelating the azimuth-elevation domain with rectangular tiles. This tessellation is unfortunately impractical since *irregular*: its polygonal cells have varying areas and shapes (large rectangular cells at the equator and elongated triangular cells near the poles). For practical purposes, regular tessellations –with equal-area and identical polygonal cells– are often preferred. The latter are however only available for fixed resolutions ($N = 4$, 6, 8, 12 and 20) and are obtained from the five Platonic solid vertices [14, Chapter 3]: the tetrahedron, cube, octahedron, dodecahedron and icosahedron. For arbitrary number of tiles, there exist many *near-regular* tilings of the sphere with almost uniform polygonal tiles [14, Chapter 3]. Among them, one counts notably the *HEALPix tessellation* [4], primarily used in cosmological applications [12]. It is obtained by hierarchically subdividing the Voronoi cells of the dodecahedron vertices into equal-area elements (see fig. B.1b). In this work, we consider moreover the *Fibonacci tessellation* [7]. Points in the Fibonacci lattice are arranged uniformly along a spiral pattern on the sphere linking the two poles (see fig. B.1c). The lattice can very easily be generated from the following formulae:

$$\begin{cases} \boldsymbol{r}_n = \left[\cos(\varphi_n)\sin(\theta_n), \sin(\varphi_n)\sin(\theta_n), \cos(\theta_n)\right], \\ \text{where} \quad \varphi_n = 2\pi n \left(1 - \frac{2}{1+\sqrt{5}}\right) \quad \& \quad \theta_n = \arccos\left(1 - \frac{2n}{N}\right), \end{cases} \quad n = 1, \ldots, N. \quad \text{(B.1)}$$

## B.2  Spherical maps as signals on graphs

Multi-dimensional arrays are particularly convenient data-structures, since their *connectivity graph* is implicitly defined[3] and preserves the notion of spatial *locality*: Euclidean distances are proportional to index offsets. Labels of spherical maps on the other hand are often arbitrary, resulting in a fundamental mismatch between connectivity in the intensity vector $\mathbf{x}$ and locality in the lattice $\Theta$. To properly account for the dependencies in $\mathbf{x}$ arising from the underlying domain geometry, one possibility [3, 12] is to define an explicit connectivity graph $\mathcal{G} = (\Theta, \mathcal{E}, \mathbf{W})$, where $\mathcal{E} \subset \Theta^2$ is an *edge set* defining neighbouring vertices in $\Theta$ and $\mathbf{W} \in \mathbb{R}^{N \times N}$ a *weighting matrix*, defining the *similarity* between two connected vertices (see fig. B.2). Given an arbitrary lattice $\Theta$, the edge set

(a) Spherical map on its connectivity graph. (b) Sparsity pattern of the associated graph Laplacian (only $\simeq$ 0.4% of nonzero values).

Figure B.2: Spherical map seen as a signal on a graph.

can for example be defined as its Delaunay triangulation, obtained from the convex-hull of $\Theta$. The edge weights are moreover commonly defined as a function of the distance separating two vertices in the lattice $\Theta$. In [12], the authors recommend the following weighting scheme:

$$[\mathbf{W}]_{nm} := \begin{cases} \exp\left(-\dfrac{\|\mathbf{r}_n - \mathbf{r}_m\|_2^2}{\rho^2}\right) & \text{if } (\mathbf{r}_n, \mathbf{r}_m) \in \mathcal{E}, \\ 0 & \text{otherwise,} \end{cases}$$

where $\rho > 0$ is given by $\rho = \frac{1}{|\mathcal{E}|}\sum_{(\mathbf{r}_n,\mathbf{r}_m)\in\mathcal{E}} \|\mathbf{r}_n - \mathbf{r}_m\|_2$. With this additional structure, a spherical map can be seen as a signal on a graph (see fig. B.2a), which can be processed by means of *graph signal processing* tools [17].

## B.3   Discrete spherical convolutions

The DeepWave RNN described in (1) performs *filtering* operations at each layer, implemented by means of *graph convolution operators* [17]. Graph convolution operators generalise Euclidean convolution operators to irregular domains such as spherical tessellations where shifts are not naturally defined. By analogy to the Euclidean setting, graph convolutions are defined as operators diagonalised by the Fourier basis, obtained from the eigenvectors of the *Laplacian* of the graph $\mathcal{G}$. As explained in [17], there exist many possible definitions of the graph Laplacian. In this paper, we proceed as in [12] and choose to work with the *normalised Laplacian* given by:

$$\mathbf{L} := \mathbf{I} - \mathbf{D}^{-1/2}\mathbf{W}\mathbf{D}^{-1/2}, \tag{B.2}$$

where $\mathbf{I} \in \mathbb{R}^{N \times N}$ denotes the *identity matrix* and $\mathbf{D} \in \mathbb{R}^{N \times N}$ is a diagonal matrix defined as:

$$[\mathbf{D}]_{ii} = \sum_{n=1}^{N} [\mathbf{W}]_{in}.$$

The Laplacian operator (B.2) has many useful properties [17]. In particular, it is often extremely sparse[4] (see fig. B.2b) and its induced norm

$$\|\mathbf{x}\|_{\mathbf{L}} := \|\mathbf{L}^{1/2}\mathbf{x}\|_2^2 = \mathbf{x}^T\mathbf{L}\mathbf{x}, \tag{B.3}$$

can be seen as a measure of *smoothness* [17, Example 2] for a signal $\mathbf{x} \in \mathbb{R}^N$ defined on the vertex set $\Theta$ of $\mathcal{G}$. If $\mathbf{L} = \mathbf{U}\boldsymbol{\Lambda}\mathbf{U}^T$ is the eigendecomposition of $\mathbf{L}$, a filter $h(\mathbf{L}) \in \mathbb{R}^{N \times N}$ is a linear operator acting on a graph signal $\mathbf{x} \in \mathbb{R}^N$ as

$$h(\mathbf{L})\mathbf{x} := \mathbf{U}h(\boldsymbol{\Lambda})\mathbf{U}^T\mathbf{x},$$

for some function $h : \mathbb{R}_+ \to \mathbb{R}$. In this work, we will consider specific graph filters for which $h$ is an order-$K$ polynomial:

$$h(\mathbf{L}) = \sum_{k=0}^{K} \theta_k \mathbf{L}^k.$$

Such filters can indeed be shown [17, 12] to have finite support in the graph domain, with radius at most $K$ vertices. Moreover, they can be efficiently implemented as a cascade of multiplications between the sparse matrix $\mathbf{L}$ and the vector $\mathbf{x}$ to be filtered. In particular, if $\mathbf{z}^0 = \mathbf{x}$ and $\mathbf{x}^0 = \theta_0 \mathbf{x}$, then the filtered vector $\tilde{\mathbf{x}}$ is given by the outcome $\mathbf{x}^K$ of the following recursion:

$$\begin{cases} \mathbf{z}^k & = \mathbf{L}\mathbf{z}^{k-1} \\ \mathbf{x}^k & = \mathbf{x}^{k-1} + \theta_k \mathbf{z}^k \end{cases} , \qquad k = 1, \dots, K. \tag{B.4}$$

For stability reasons, we consider in practice an equivalent version of (B.4), as recommended in [17, 12]:

$$\begin{cases} \mathbf{z}^k & = 2\tilde{\mathbf{L}}\mathbf{z}^{k-1} - \mathbf{z}^{k-2} \\ \mathbf{x}^k & = \mathbf{x}^{k-1} + \tilde{\theta}_k \mathbf{z}^k \end{cases} , \qquad k = 2, \dots, K, \tag{B.5}$$

with $\mathbf{x}^0 = \mathbf{z}^0 = \mathbf{x}$, $\mathbf{z}^1 = \tilde{\mathbf{L}}\mathbf{x}$ and $\mathbf{x}^1 = \tilde{\theta}_1 \mathbf{z}^1 + \tilde{\theta}_2 \mathbf{z}^2$. The weights $\{\tilde{\theta}_0, \dots, \tilde{\theta}_K\} \subset \mathbb{R}$ in (B.5) are such that

$$\sum_{k=0}^{K} \theta_k \mathbf{L}^k = \sum_{k=0}^{K} \tilde{\theta}_k T_k(\tilde{\mathbf{L}}),$$

where $T_k : [-1, 1] \to \mathbb{R}$ are *Chebyshev polynomials* and $\tilde{\mathbf{L}}$ is the Laplacian with rescaled and shifted spectrum in the interval $[-1, 1]$ [12]:

$$\tilde{\mathbf{L}} = \frac{2}{\lambda_{max}}\mathbf{L} - \mathbf{I}.$$

Note that eq. (B.5) is obtained from the recursion formula defining the Chebyshev polynomials: $T_k(x) = 2xT_{k-1}(x) - T_{k-2}(x)$, with $T_1(x) = x$ and $T_0(x) = 1$.

## C  Background: far-field point source reconstruction problem 2

This section provides background material on classical objective function (2).

Narrow-band far-field acoustic perturbations are modeled in baseband-equivalent form [19] as realisations of a random function

$$\mathcal{S} = \left\{ S(\mathbf{r}) : \Omega \to \mathbb{C}, \ \mathbf{r} \in \mathbb{S}^2 \right\},$$

where $S(\mathbf{r})$ follows a zero-mean, spatially-uncorrelated complex Gaussian distribution of variance (i.e. intensity) $x(\mathbf{r}) = \mathbb{E}\left[S(\mathbf{r})S^*(\mathbf{r})\right] \in \mathbb{R}_+$. Concretely, when mapped onto a discrete spherical tesselation of resoluton $N$, $S(\mathbf{r})$ is given by

$$S(\mathbf{r}) = \sum_{q=1}^{N} \xi_q \delta(\mathbf{r} - \mathbf{r}_q), \tag{C.1}$$

where $\xi_q \sim \mathbb{C}\mathcal{N}(0, x_q)$ and $\Theta = \{\mathbf{r}_1, \dots, \mathbf{r}_N\} \subset \mathbb{S}^2$ are the tesselation support points. The goal in the acoustic imaging problem is to estimate the intensity field $\mathbf{x} = [x_1, \dots, x_N] \in \mathbb{R}_+^N$.

Knowledge of $\mathbf{x}$ can be gathered by observing realisations of $\mathcal{S}$ using a microphone array. Let $\mathbf{y} : \Omega \to \mathbb{C}^M$ denote the random samples measured at the output of an $M$-element array. Then microphone samples are linked to $\mathcal{S}$ through the relation [20]

$$[\mathbf{y}]_m = \int_{\mathbb{S}^2} S(\mathbf{r}) \exp\left(-j\frac{2\pi}{\lambda_0}\langle \mathbf{r}, \mathbf{p}_m \rangle\right) d\mathbf{r} \overset{(\text{C.1})}{=} \sum_{q=1}^{N} \xi_q \exp\left(-j\frac{2\pi}{\lambda_0}\langle \mathbf{r}_q, \mathbf{p}_m \rangle\right), \tag{C.2}$$

where $\mathbf{p}_m \in \mathbb{R}^3$ denotes the position of the $m$-th microphone, and $\lambda_0 > 0$ is the wavelength of the impeding plane wave. The covariance matrix $\boldsymbol{\Sigma} \in \mathbb{C}^{M \times M}$ then directly links $\mathbf{x}$ to the measurements as

$$[\boldsymbol{\Sigma}]_{ij} = \mathbb{E}\left[[\mathbf{y}]_i [\mathbf{y}]_j^*\right] = \sum_{q=1}^{N} \sum_{l=1}^{N} \mathbb{E}\left[\xi_q \xi_l^*\right] \exp\left(-j\frac{2\pi}{\lambda_0}\langle \mathbf{r}_q, \mathbf{p}_i\rangle\right) \exp\left(j\frac{2\pi}{\lambda_0}\langle \mathbf{r}_l, \mathbf{p}_j\rangle\right)$$

$$= \sum_{q=1}^{N} [\mathbf{x}]_q \exp\left(j\frac{2\pi}{\lambda_0}\langle \mathbf{r}_q, \mathbf{p}_j - \mathbf{p}_i\rangle\right),$$

which can be succintely described by matrix equation

$$\boldsymbol{\Sigma} = \mathbf{A}\operatorname{diag}(\mathbf{x})\mathbf{A}^H, \quad \mathbf{A} = \exp\left(-j\frac{2\pi}{\lambda_0}\mathbf{P}^T\mathbf{R}\right) \in \mathbb{C}^{M \times N}, \tag{C.3}$$

where $\mathbf{P} = [\mathbf{p}_1, \ldots, \mathbf{p}_M] \in \mathbb{R}^{3 \times M}$ and $\mathbf{R} = [\mathbf{r}_1, \ldots, \mathbf{r}_N] \in \mathbb{R}^{3 \times N}$. Replacing $\boldsymbol{\Sigma}$ in (C.3) by the empirical covariance matrix $\hat{\boldsymbol{\Sigma}} \in \mathbb{C}^{M \times M}$ then gives rise to the data-fidelity term

$$\frac{1}{2}\left\|\hat{\boldsymbol{\Sigma}} - \mathbf{A}\operatorname{diag}(\mathbf{x})\mathbf{A}^H\right\|_F^2. \tag{C.4}$$

Minimisation of (C.4) alone is not enough to obtain a unique minimiser due to limited data availability, i.e. $\binom{M}{2} \ll N$. To overcome this limitation and allow group-sparse reconstructions characteristic of acoustic scenes, (C.4) is augmented with the elastic-net regulariser [25]

$$\lambda\left[\gamma\|\mathbf{x}\|_1 + (1-\gamma)\|\mathbf{x}\|_2^2\right]. \tag{C.5}$$

Combining (C.4) and (C.5) leads to (2).

# D   Derivation: proximal gradient descent for elastic-net problem 3

This section shows how to obtain proximal iteration (4) from (3).

Recall that the sound intensity map is obtained by solving the convex optimisation problem:

$$\hat{\mathbf{x}} = \operatorname*{arg\,min}_{\mathbf{x} \in \mathbb{R}_+^N} f(\mathbf{x}) + g(\mathbf{x}), \tag{D.1}$$

$$f(\mathbf{x}) = \frac{1}{2}\left\|\hat{\boldsymbol{\Sigma}} - \mathbf{A}\operatorname{diag}(\mathbf{x})\mathbf{A}^H\right\|_F^2 \overset{(A.5)}{=} \frac{1}{2}\left\|\operatorname{vec}(\hat{\boldsymbol{\Sigma}}) - \left(\overline{\mathbf{A}} \circ \mathbf{A}\right)\mathbf{x}\right\|_2^2, \tag{D.2}$$

$$g(\mathbf{x}) = \lambda\left[\gamma\|\mathbf{x}\|_1 + (1-\gamma)\|\mathbf{x}\|_2^2\right], \tag{D.3}$$

where $g$ is an elastic-net regularizer with $\lambda \geq 0$ and $\gamma \in ]0, 1[$.

Proximal gradient descent (PGD) is a fixed-point method to solve problems of the form (D.1) where $f$, $g$ are closed proper convex with $f$ differentiable. It consists of iterating the proximal update equation until convergence:

$$\mathbf{x}^k = \operatorname{prox}_{\alpha g}\left(\mathbf{x}^{k-1} - \alpha\nabla f(\mathbf{x}^{k-1})\right), \tag{D.4}$$

where $\alpha > 0$ is the step size and $\operatorname{prox}_{\alpha g}$ is the proximal operator associated with (D.3), given by (see proof below):

$$\operatorname{prox}_{\alpha g}(\mathbf{x}) = \operatorname*{arg\,min}_{\mathbf{u} \in \mathbb{R}_+^N} g(\mathbf{u}) + \frac{1}{2\alpha}\|\mathbf{u} - \mathbf{x}\|_2^2, \tag{D.5}$$

$$= \operatorname{ReLu}\left(\frac{\mathbf{x} - \lambda\alpha\gamma}{2\lambda\alpha(1-\gamma) + 1}\right), \quad \forall\mathbf{x} \in \mathbb{R}^N. \tag{D.6}$$

The quantity $\nabla f \in \mathbb{R}^N$ finally is obtained using the rules of vector calculus [13]:

$$\nabla f(\mathbf{x}) = \left\{\frac{\partial}{\partial\mathbf{x}}\left[\operatorname{vec}(\hat{\boldsymbol{\Sigma}}) - \left(\overline{\mathbf{A}} \circ \mathbf{A}\right)\mathbf{x}\right]\right\} \cdot \left[\operatorname{vec}(\hat{\boldsymbol{\Sigma}}) - \left(\overline{\mathbf{A}} \circ \mathbf{A}\right)\mathbf{x}\right]$$

$$= \left(\overline{\mathbf{A}} \circ \mathbf{A}\right)^H\left[\left(\overline{\mathbf{A}} \circ \mathbf{A}\right)\mathbf{x} - \operatorname{vec}(\hat{\boldsymbol{\Sigma}})\right]. \tag{D.7}$$

Combining (D.4), (D.6) and (D.7) leads to (4).

*Proof: (Analytic expression for* $\text{prox}_{\alpha g}$*).* Replacing (D.3) in (D.5), we get for $\mathbf{x} \in \mathbb{R}^N$:

$$\text{prox}_{\alpha g}(\mathbf{x}) = \underset{\mathbf{u} \in \mathbb{R}_+^N}{\arg\min} \, \lambda \left[ \gamma \|\mathbf{u}\|_1 + (1 - \gamma) \|\mathbf{u}\|_2^2 \right] + \frac{1}{2\alpha} \|\mathbf{u} - \mathbf{x}\|_2^2$$

$$= \underset{(u_1,\ldots,u_N) \in \mathbb{R}_+^N}{\arg\min} \sum_{n=1}^N \lambda \left[ \gamma |u_n| + (1 - \gamma)u_n^2 \right] + \frac{1}{2\alpha}(u_n - x_n)^2$$

$$= \underset{(u_1,\ldots,u_N) \in \mathbb{R}_+^N}{\arg\min} \sum_{n=1}^N \lambda \left[ \gamma u_n + (1 - \gamma)u_n^2 \right] + \frac{1}{2\alpha} \left[ u_n^2 + x_n^2 - 2u_n x_n \right]$$

$$= \underset{(u_1,\ldots,u_N) \in \mathbb{R}_+^N}{\arg\min} \sum_{n=1}^N \varphi_n(u_n). \tag{D.8}$$

Notice that (D.8) is the sum of $N$ independent objective functionals, hence each can be independently minimised. (We drop the subscript of $\varphi_n$ below for simplicity.) Let $\hat{u}$ be the minimiser:[5]

$$\hat{u} = \underset{u \geq 0}{\arg\min} \, \varphi(u) = \underset{u \geq 0}{\arg\min} \, \lambda \left[ \gamma u + (1 - \gamma)u^2 \right] + \frac{1}{2\alpha} \left[ u^2 + x^2 - 2ux \right], \tag{D.9}$$

for some fixed $x \in \mathbb{R}$. Then two cases can occur:

- $x \leq 0$: the objective functional being composed of positive terms only, any $\hat{u} > 0$ will increase the objective. Therefore $\hat{u} = 0$.

- $x > 0$: In this case the Karush Kuhn Tucker (KKT) conditions [18, 2] tell us that $\hat{u}$ is a minimizer of (D.9) if

$$\hat{u}\varphi'(\hat{u}) = 0$$
$$\varphi'(\hat{u}) \geq 0 \quad \text{if } \hat{u} = 0.$$

Plugging $\varphi'(u) = \lambda\gamma + \left( 2\lambda(1 - \gamma) + \alpha^{-1} \right) u - \alpha^{-1}x$ and solving the above yields

$$\hat{u} = \begin{cases} \frac{x - \lambda\alpha\gamma}{2\lambda\alpha(1-\gamma)+1} & x > \lambda\alpha\gamma, \\ 0 & x \leq \lambda\alpha\gamma \end{cases}.$$

Both cases can be written in short as

$$\hat{u} = \underset{u \geq 0}{\arg\min} \, \varphi(u) = \left[ \frac{x - \lambda\alpha\gamma}{2\lambda\alpha(1 - \gamma) + 1} \right]_+, \qquad \forall x \in \mathbb{R},$$

leading to an element-wise proximal operator of the form

$$\text{prox}_{\alpha g}(\mathbf{x}) = \left[ \frac{\mathbf{x} - \lambda\alpha\gamma}{2\lambda\alpha(1 - \gamma) + 1} \right]_+ = \text{ReLu}\left( \frac{\mathbf{x} - \lambda\alpha\gamma}{2\lambda\alpha(1 - \gamma) + 1} \right), \qquad \forall \mathbf{x} \in \mathbb{R}^N.$$

$\square$

# E  Proof: proposition 1

In this section, we prove proposition 1 of the main paper:

**Proposition.** *Consider a* spherical microphone array, *with diameter D and microphone directions* $\{\tilde{\mathbf{p}}_1, \ldots, \tilde{\mathbf{p}}_M\} \subset \mathbb{S}^2$, *forming a* near-regular tessellation *of the sphere. Then, we have*

$$\left[ I - \alpha \left( \overline{\mathbf{A}} \circ \mathbf{A} \right)^H \left( \overline{\mathbf{A}} \circ \mathbf{A} \right) \right]_{ij} \simeq \left[ \delta_{ij} - \alpha M^2 \, \text{sinc}^2 \left( \frac{D}{\lambda_0} \|\mathbf{r}_i - \mathbf{r}_j\| \right) \right], \, \forall i, j \in \{1, \ldots, N\} \tag{E.1}$$

*where* $\lambda_0$ *is the* wavelength, $\delta_{ij}$ *denotes the* Kronecker delta *and* $\text{sinc}(x) := \sin(\pi x)/\pi x$ *is the* cardinal sine. *Moreover, the approximation* (E.1) *is* extremely good *for* $M \geq 3 \lfloor \frac{2\pi D}{\lambda_0} \rfloor^2$.

(a) Beamshape of the Pyramic array.

(b) Approximate beamshape obtained with (E.2).

Figure E.1: Accuracy of approximation (E.2) for the Pyramic array [16] ($D = 30$[cm], $M = 48$) at 1 kHz.

*Proof.* To prove (E.1), it is sufficient to show that

$$\left[ \left( \overline{\mathbf{A}} \circ \mathbf{A} \right)^H \left( \overline{\mathbf{A}} \circ \mathbf{A} \right) \right]_{ij} \simeq M^2 \operatorname{sinc}^2 \left( \frac{D}{\lambda_0} \| \mathbf{r}_i - \mathbf{r}_j \| \right). \tag{E.2}$$

To this end, we first use (A.11) and obtain:

$$\left( \overline{\mathbf{A}} \circ \mathbf{A} \right)^H \left( \overline{\mathbf{A}} \circ \mathbf{A} \right) = \left| \mathbf{A}^H \mathbf{A} \right|^{\odot 2}. \tag{E.3}$$

For a spherical array with diameter $D$ and microphone directions $\{ \tilde{\mathbf{p}}_1, \ldots, \tilde{\mathbf{p}}_M \} \subset \mathbb{S}^2$, we get moreover from the definition of the steering matrix that:

$$[\mathbf{A}^H \mathbf{A}]_{ij} = \sum_{m=1}^{M} \exp \left( j \frac{\pi D}{\lambda_0} \langle \mathbf{r}_i - \mathbf{r}_j, \tilde{\mathbf{p}}_m \rangle \right), \qquad i, j = 1, \ldots, N. \tag{E.4}$$

Since the microphone directions are assumed to form a near-regular tessellation over the sphere (such as the Fibonacci or HEALPix tessellations discussed in appendix B.1), we can interpret (E.4) as a quadrature rule on the sphere, yielding:

$$\frac{4\pi}{M} \sum_{m=1}^{M} \exp \left( j \frac{\pi D}{\lambda_0} \langle \mathbf{r}_i - \mathbf{r}_j, \tilde{\mathbf{p}}_m \rangle \right) \simeq \int_{\mathbb{S}^2} \exp \left( j \frac{\pi D}{\lambda_0} \langle \mathbf{r}_i - \mathbf{r}_j, \tilde{\mathbf{p}} \rangle \right) d\tilde{\mathbf{p}} \tag{E.5}$$

$$= 4\pi \operatorname{sinc} \left( \frac{D}{\lambda_0} \| \mathbf{r}_i - \mathbf{r}_j \| \right), \tag{E.6}$$

where the second equality (E.6) follows from the result on [21, p. 154]. From (E.6), (E.4) and (E.3) we obtain (E.2) from which (E.1) trivially follows.

Regarding the quality of the approximation (E.2) finally, we use the approximate bandlimitedness of complex plane-waves in the spherical domain [14, Chapter 2]. Indeed, quadrature rules such as

(E.5) are almost exact for bandlimited functions [14, Chapter 3], provided a high-enough number of quadrature points $M$. For example, a function with spherical harmonic bandwidth $L \in \mathbb{N}$ is extremely well approximated by the HEALPix quadrature rule for $M \geq 3L^2$ [4]. In our case, the *plane-wave expansion* [14, Chapter 2] gives us

$$\exp\left(j\frac{\pi D}{\lambda_0}\langle \mathbf{r}_i - \mathbf{r}_j, \tilde{\mathbf{p}}\rangle\right) = 4\pi \sum_{l=0}^{+\infty} \sum_{k=-l}^{l} j^l(2l+1)j_l\left(\frac{\pi D}{\lambda_0}\|\mathbf{r}_i - \mathbf{r}_j\|_2\right)\overline{Y_l^k}(\tilde{\mathbf{r}}_{ij})Y_l^k(\tilde{\mathbf{p}}),$$

where $j_l$ are *spherical Bessel functions*, $Y_l^k$ *spherical harmonics*, and $\tilde{\mathbf{r}}_{ij} = (\mathbf{r}_i - \mathbf{r}_j)/\|\mathbf{r}_i - \mathbf{r}_j\|_2^2$ [14]. Since $j_l(x) \simeq 0$ for $l \geq x$ [14, Chapter 2] we have hence that complex plane-waves are approximately bandlimited with bandwidth $L = \lfloor \frac{\pi D}{\lambda_0}\|\mathbf{r}_i - \mathbf{r}_j\|_2 \rfloor \leq \lfloor \frac{2\pi D}{\lambda_0} \rfloor$. As a result, choosing $M \geq 3\lfloor \frac{2\pi D}{\lambda_0} \rfloor^2$ makes the approximation (E.2) very accurate. $\qquad\square$

While proven for spherical arrays only, approximation (E.2) (and hence (E.1)) remains quite accurate in practice, even for non spherical microphone arrays such as the Pyramic array used in our real-world experiments [16]. In fig. E.1, we investigated visually the quality of the approximation (E.2) for the Pyramic array at 1 kHZ. To this end, we plotted a row of $\left|\mathbf{A}^H \mathbf{A}\right|^{\odot 2}$ (which corresponds to the beamshape of the instrument for a particular direction [23]) with and without approximation. We observe that the approximation is already very good, even if the Pyramic array possesses only $M = 48$ microphones against the 90 required by proposition 1 for an optimal approximation accuracy at this frequency.

# F   Network gradient evaluation

This section shows how to obtain derivatives of data-fidelity term $\mathcal{L}_t$ from eq. (9) w.r.t. network parameters $\boldsymbol{\theta}, \mathbf{B}, \boldsymbol{\tau}$.[6]

## F.1   Problem statement

Recall that

$$\nabla\mathcal{L}(\boldsymbol{\Omega}) = \left\{\frac{\partial\mathcal{L}}{\partial\boldsymbol{\theta}} \in \mathbb{R}^{K+1}, \ \frac{\partial\mathcal{L}}{\partial\mathbf{B}} \in \mathbb{C}^{M\times N}, \ \frac{\partial\mathcal{L}}{\partial\boldsymbol{\tau}} \in \mathbb{R}^N\right\},$$

$$\mathcal{L}(\boldsymbol{\Omega}) = \frac{1}{2}\frac{\left\|\hat{\mathbf{x}} - \mathbf{x}^L(\boldsymbol{\Omega})\right\|_2^2}{\|\hat{\mathbf{x}}\|_2^2}, \tag{F.1}$$

where $\mathbf{x}^L(\boldsymbol{\Omega}) \in \mathbb{R}_+^N$ is given by recurrence relation (1):

$$\mathbf{x}^l(\boldsymbol{\Omega}) = \sigma\left[P_{\boldsymbol{\theta}}(\mathbf{L})\mathbf{x}^{l-1} + \left(\overline{\mathbf{B}} \circ \mathbf{B}\right)^H \text{vec}(\hat{\boldsymbol{\Sigma}}) - \boldsymbol{\tau}\right] \tag{F.2}$$

$$= \sigma\left[\mathbf{u}^l + \mathbf{w} - \boldsymbol{\tau}\right] \tag{F.3}$$

$$= \sigma\left[\mathbf{s}^l\right], \qquad l = 1, \dots, L \tag{F.4}$$

with $\mathbf{x}^0 \in \mathbb{R}_+^N$ some arbitrary constant, $\sigma : \mathbb{R} \to \mathbb{R}$ a point-wise non-linearity, and $P_{\boldsymbol{\theta}}(\mathbf{L}) = \sum_{k=0}^K \theta_k T_k(\mathbf{L})$ a polynomial filter of order $K$ expressed in terms of Chebychev polynomials.

$\nabla\mathcal{L}$ can be efficiently evaluated using *reverse-mode algorithmic differentiation*[1, 6] in a two-stage process:

- Forward pass: evaluate eq. (F.1) while storing all intermediate values $\mathbf{w}, \boldsymbol{\tau}, \left\{\mathbf{s}^l\right\}_{l=1,\dots,L}$;

- Backward pass: walk the computational graph (fig. F.1) backwards to evaluate derivatives w.r.t. $\boldsymbol{\theta}, \mathbf{B}, \boldsymbol{\tau}$.

Figure F.1: $L$-layer computational graph of $\mathcal{L}$.

## F.2 Conventions

- If $\mathbf{u} \in \mathbb{R}^N$, $\mathbf{v} \in \mathbb{R}^M$, the *Jacobian matrix* $\frac{\partial \mathbf{u}}{\partial \mathbf{v}} \in \mathbb{R}^{N \times M}$ is defined as

$$\left[ \frac{\partial \mathbf{u}}{\partial \mathbf{v}} \right]_{ij} = \frac{\partial [\mathbf{u}]_i}{\partial [\mathbf{v}]_j}.$$

Gradients of scalar-valued functions are therefore row vectors.

- If $\mathbf{u} \in \mathbb{R}^N$, $\mathbf{V} \in \mathbb{R}^{M \times Q}$, the *Jacobian tensor* $\frac{\partial \mathbf{u}}{\partial \mathbf{V}} \in \mathbb{R}^{N \times M \times Q}$ is defined as

$$\left[ \frac{\partial \mathbf{u}}{\partial \mathbf{V}} \right]_{ijk} = \frac{\partial [\mathbf{u}]_i}{\partial [\mathbf{V}]_{jk}}.$$

## F.3 Common intermediate gradients

$$\left[ \frac{\partial \mathcal{L}}{\partial \mathbf{x}^L} \right]_i = \frac{\partial \mathcal{L}}{\partial [\mathbf{x}^L]_i} = \left[ \frac{\mathbf{x}^L - \hat{\mathbf{x}}}{\|\hat{\mathbf{x}}\|_2^2} \right]_i \tag{F.5}$$

$$\left[ \frac{\partial \mathbf{x}^l}{\partial \mathbf{s}^l} \right]_{ij} = \frac{\partial [\mathbf{x}^l]_i}{\partial [\mathbf{s}^l]_j} = \delta_{i-j} \sigma' \left( [\mathbf{s}^l]_j \right) = \left[ \operatorname{diag} \left( \sigma' \left( \mathbf{s}^l \right) \right) \right]_{ij}, \qquad l = 1, \dots, L \tag{F.6}$$

$$\frac{\partial \mathcal{L}}{\partial \mathbf{s}^l} = \frac{\partial \mathcal{L}}{\partial \mathbf{x}^l} \frac{\partial \mathbf{x}^l}{\partial \mathbf{s}^l} \overset{\text{(F.6)}}{=} \frac{\partial \mathcal{L}}{\partial \mathbf{x}^l} \operatorname{diag} \left( \sigma' \left( \mathbf{s}^l \right) \right), \qquad l = 1, \dots, L \tag{F.7}$$

$$\left[ \frac{\partial \mathbf{s}^l}{\partial \mathbf{u}^l} \right]_{ij} = \frac{\partial [\mathbf{s}^l]_i}{\partial [\mathbf{u}^l]_j} = \frac{\partial}{\partial [\mathbf{u}^l]_j} \left[ \mathbf{u}^l + \mathbf{w} - \boldsymbol{\tau} \right]_i = \delta_{i-j} = [\mathbf{I}_N]_{ij} \tag{F.8}$$

$$\frac{\partial \mathcal{L}}{\partial \mathbf{u}^l} = \frac{\partial \mathcal{L}}{\partial \mathbf{s}^l} \frac{\partial \mathbf{s}^l}{\partial \mathbf{u}^l} \overset{\text{(F.8)}}{=} \frac{\partial \mathcal{L}}{\partial \mathbf{s}^l}, \qquad l = 1, \dots, L \tag{F.9}$$

$$\left[ \frac{\partial \mathbf{u}^l}{\partial \mathbf{x}^{l-1}} \right]_{ij} = \left[ \frac{\partial}{\partial \mathbf{x}^{l-1}} P_{\boldsymbol{\theta}}(\mathbf{L}) \mathbf{x}^{l-1} \right]_{ij} = [P_{\boldsymbol{\theta}}(\mathbf{L})]_{ij}, \qquad l = 1, \dots, L \tag{F.10}$$

$$\left[ \frac{\partial \mathbf{s}^l}{\partial \mathbf{w}} \right]_{ij} = \frac{\partial [\mathbf{s}^l]_i}{\partial [\mathbf{w}]_j} = \frac{\partial}{\partial [\mathbf{w}]_j} \left[ \mathbf{u}^l + \mathbf{w} - \boldsymbol{\tau} \right]_i = \delta_{i-j} = [\mathbf{I}_N]_{ij} \tag{F.11}$$

$$\frac{\partial \mathcal{L}}{\partial \mathbf{w}} = \sum_{l=1}^{L} \frac{\partial \mathcal{L}}{\partial \mathbf{s}^l} \frac{\partial \mathbf{s}^l}{\partial \mathbf{w}} \overset{\text{(F.11)}}{=} \sum_{l=1}^{L} \frac{\partial \mathcal{L}}{\partial \mathbf{s}^l} \tag{F.12}$$

## F.4 $\partial\mathcal{L}/\partial\boldsymbol{\theta}$

$$\left[\frac{\partial\mathbf{u}^l}{\partial\boldsymbol{\theta}}\right]_{ij} = \frac{\partial}{\partial\left[\boldsymbol{\theta}\right]_j}\sum_{k=0}^{K}\left[\boldsymbol{\theta}\right]_k\left[T_k(\mathbf{L})\mathbf{x}^{l-1}\right]_i = \left[T_j(\mathbf{L})\mathbf{x}^{l-1}\right]_i, \qquad l = 1,\ldots,L \tag{F.13}$$

$$\left[\frac{\partial\mathcal{L}}{\partial\boldsymbol{\theta}}\right]_i = \sum_{l=1}^{L}\left[\frac{\partial\mathcal{L}}{\partial\mathbf{u}^l}\frac{\partial\mathbf{u}^l}{\partial\boldsymbol{\theta}}\right]_i \overset{(\text{F.9})}{\underset{(\text{F.13})}{=}} \sum_{i=1}^{L}\frac{\partial\mathcal{L}}{\partial\mathbf{s}^l}T_i(\mathbf{L})\mathbf{x}^{l-1}, \qquad i = 0,\ldots,K \tag{F.14}$$

## F.5 $\partial\mathcal{L}/\partial\mathbf{B}$

$\frac{\partial\mathcal{L}}{\partial\mathbf{B}}$ can be obtained by evaluating $\frac{\partial\mathcal{L}}{\partial\mathbf{w}}\frac{\partial\mathbf{w}}{\partial\mathbf{B}}$, but $\frac{\partial\mathbf{w}}{\partial\mathbf{B}} \in \mathbb{C}^{N\times M\times N}$ is difficult to obtain directly. We therefore proceed in multiple steps:

1. Decompose $\mathbf{w}$ as $(\mathbf{w}_1 + \mathbf{w}_2 + \mathbf{w}_3)$ and express $\{\mathbf{w}_k\}_{k=1,2,3}$ explicitly in terms of $\hat{\boldsymbol{\Sigma}}_R, \hat{\boldsymbol{\Sigma}}_I, \mathbf{B}_R, \mathbf{B}_I$:

$$\begin{aligned}
\mathbf{w} &= \left(\overline{\mathbf{B}}\circ\mathbf{B}\right)^H \text{vec}(\hat{\boldsymbol{\Sigma}}) \overset{(\text{A.8})}{=} \text{diag}\left(\mathbf{B}^H\hat{\boldsymbol{\Sigma}}\mathbf{B}\right) \\
&= \text{diag}\left(\left[\mathbf{B}_R + j\mathbf{B}_I\right]^H\left[\hat{\boldsymbol{\Sigma}}_R + j\hat{\boldsymbol{\Sigma}}_I\right]\left[\mathbf{B}_R + j\mathbf{B}_I\right]\right) \\
&= \text{diag}\left(\left[\mathbf{B}_R^T - j\mathbf{B}_I^T\right]\left[\hat{\boldsymbol{\Sigma}}_R + j\hat{\boldsymbol{\Sigma}}_I\right]\left[\mathbf{B}_R + j\mathbf{B}_I\right]\right) \\
&= \text{diag}\left(\mathbf{B}_R^T\hat{\boldsymbol{\Sigma}}_R\mathbf{B}_R + \mathbf{B}_I^T\hat{\boldsymbol{\Sigma}}_R\mathbf{B}_I + \mathbf{B}_I^T\hat{\boldsymbol{\Sigma}}_I\mathbf{B}_R - \mathbf{B}_R^T\hat{\boldsymbol{\Sigma}}_I\mathbf{B}_I\right) \\
&\quad + j\,\text{diag}\left(\mathbf{B}_R^T\hat{\boldsymbol{\Sigma}}_I\mathbf{B}_R + \mathbf{B}_I^T\hat{\boldsymbol{\Sigma}}_I\mathbf{B}_I + \mathbf{B}_R^T\hat{\boldsymbol{\Sigma}}_R\mathbf{B}_I - \mathbf{B}_I^T\hat{\boldsymbol{\Sigma}}_R\mathbf{B}_R\right) \\
&\overset{\mathbf{w}\in\mathbb{R}_+^N}{=} \text{diag}\left(\mathbf{B}_R^T\hat{\boldsymbol{\Sigma}}_R\mathbf{B}_R + \mathbf{B}_I^T\hat{\boldsymbol{\Sigma}}_R\mathbf{B}_I + \mathbf{B}_I^T\hat{\boldsymbol{\Sigma}}_I\mathbf{B}_R\right) - \text{diag}\left(\mathbf{B}_R^T\hat{\boldsymbol{\Sigma}}_I\mathbf{B}_I\right) \\
&= \text{diag}\left(\mathbf{B}_R^T\hat{\boldsymbol{\Sigma}}_R\mathbf{B}_R + \mathbf{B}_I^T\hat{\boldsymbol{\Sigma}}_R\mathbf{B}_I + \mathbf{B}_I^T\hat{\boldsymbol{\Sigma}}_I\mathbf{B}_R\right) - \text{diag}\left(\mathbf{B}_I^T\hat{\boldsymbol{\Sigma}}_I^T\mathbf{B}_R\right) \\
&\overset{\hat{\boldsymbol{\Sigma}}_I=-\hat{\boldsymbol{\Sigma}}_I^T}{=} \text{diag}\left(\mathbf{B}_R^T\hat{\boldsymbol{\Sigma}}_R\mathbf{B}_R + \mathbf{B}_I^T\hat{\boldsymbol{\Sigma}}_R\mathbf{B}_I + \mathbf{B}_I^T\hat{\boldsymbol{\Sigma}}_I\mathbf{B}_R\right) + \text{diag}\left(\mathbf{B}_I^T\hat{\boldsymbol{\Sigma}}_I\mathbf{B}_R\right) \\
&= \text{diag}\left(\mathbf{B}_R^T\hat{\boldsymbol{\Sigma}}_R\mathbf{B}_R + \mathbf{B}_I^T\hat{\boldsymbol{\Sigma}}_R\mathbf{B}_I + 2\mathbf{B}_I^T\hat{\boldsymbol{\Sigma}}_I\mathbf{B}_R\right) \\
&\overset{(\text{A.8})}{=} \underbrace{\left(\mathbf{B}_R\circ\mathbf{B}_R\right)^T\text{vec}(\hat{\boldsymbol{\Sigma}}_R)}_{\mathbf{w}_1} + \underbrace{\left(\mathbf{B}_I\circ\mathbf{B}_I\right)^T\text{vec}(\hat{\boldsymbol{\Sigma}}_R)}_{\mathbf{w}_2} + \underbrace{2\left(\mathbf{B}_R\circ\mathbf{B}_I\right)^T\text{vec}(\hat{\boldsymbol{\Sigma}}_I)}_{\mathbf{w}_3}.
\end{aligned} \tag{F.15}$$

2. Derive analytic forms for $\left\{\frac{\partial\mathbf{w}_k}{\partial\mathbf{B}_{R/I}}\right\}_{k=1,2,3}$:

$$\begin{aligned}
\left[\frac{\partial\mathbf{w}_1}{\partial\mathbf{B}_R}\right]_{ijk} &= \frac{\partial\left[\mathbf{w}_1\right]_i}{\partial\left[\mathbf{B}_R\right]_{jk}} \overset{(\text{A.8})}{=} \frac{\partial}{\partial\left[\mathbf{B}_R\right]_{jk}}\left[\text{diag}\left(\mathbf{B}_R^T\hat{\boldsymbol{\Sigma}}_R\mathbf{B}_R\right)\right]_i \\
&= \frac{\partial}{\partial\left[\mathbf{B}_R\right]_{jk}}(\mathbf{b}_i^R)^T\hat{\boldsymbol{\Sigma}}_R\mathbf{b}_i^R \\
&= \delta_{i-k}\frac{\partial}{\partial\left[\mathbf{B}_R\right]_{jk}}(\mathbf{b}_k^R)^T\hat{\boldsymbol{\Sigma}}_R\mathbf{b}_k^R \\
&= \delta_{i-k}\sum_{q=1}^{M}\sum_{g=1}^{M}\left[\hat{\boldsymbol{\Sigma}}_R\right]_{qg}\frac{\partial}{\partial\left[\mathbf{B}_R\right]_{jk}}\left\{\left[\mathbf{B}_R\right]_{qk}\left[\mathbf{B}_R\right]_{gk}\right\} \\
&= \delta_{i-k}\sum_{q=1}^{M}\left(\left[\hat{\boldsymbol{\Sigma}}_R\right]_{jq} + \left[\hat{\boldsymbol{\Sigma}}_R\right]_{qj}\right)\left[\mathbf{B}_R\right]_{qk} \\
&\overset{\hat{\boldsymbol{\Sigma}}_R=\hat{\boldsymbol{\Sigma}}_R^T}{=} 2\delta_{i-k}(\mathbf{b}_k^R)^T\sigma_j^R
\end{aligned} \tag{F.16}$$

$$\left[\frac{\partial \mathbf{w}_2}{\partial \mathbf{B}_I}\right]_{ijk} = \frac{\partial \left[\mathbf{w}_2\right]_i}{\partial \left[\mathbf{B}_I\right]_{jk}} \overset{\text{(A.8)}}{=} \frac{\partial}{\partial \left[\mathbf{B}_I\right]_{jk}} \left[\text{diag}\left(\mathbf{B}_I^T \hat{\mathbf{\Sigma}}_R \mathbf{B}_I\right)\right]_i \tag{F.17}$$

$$= \frac{\partial}{\partial \left[\mathbf{B}_I\right]_{jk}} (\mathbf{b}_i^I)^T \hat{\mathbf{\Sigma}}_R \mathbf{b}_i^I$$

$$= \delta_{i-k} \frac{\partial}{\partial \left[\mathbf{B}_I\right]_{jk}} (\mathbf{b}_k^I)^T \hat{\mathbf{\Sigma}}_R \mathbf{b}_k^I$$

$$= \delta_{i-k} \sum_{q=1}^{M} \sum_{g=1}^{M} \left[\hat{\mathbf{\Sigma}}_R\right]_{qg} \frac{\partial}{\partial \left[\mathbf{B}_I\right]_{jk}} \left\{\left[\mathbf{B}_I\right]_{qk} \left[\mathbf{B}_I\right]_{gk}\right\}$$

$$= \delta_{i-k} \sum_{q=1}^{M} \left(\left[\hat{\mathbf{\Sigma}}_R\right]_{jq} + \left[\hat{\mathbf{\Sigma}}_R\right]_{qj}\right) \left[\mathbf{B}_I\right]_{qk}$$

$$\overset{\hat{\mathbf{\Sigma}}_R = \hat{\mathbf{\Sigma}}_R^T}{=} 2\delta_{i-k} (\mathbf{b}_k^I)^T \sigma_j^R$$

$$\left[\frac{\partial \mathbf{w}_3}{\partial \mathbf{B}_R}\right]_{ijk} = \frac{\partial \left[\mathbf{w}_3\right]_i}{\partial \left[\mathbf{B}_R\right]_{jk}} \overset{\text{(A.8)}}{=} 2\frac{\partial}{\partial \left[\mathbf{B}_R\right]_{jk}} \left[\text{diag}\left(\mathbf{B}_I^T \hat{\mathbf{\Sigma}}_I \mathbf{B}_R\right)\right]_i \tag{F.18}$$

$$= 2\frac{\partial}{\partial \left[\mathbf{B}_R\right]_{jk}} (\mathbf{b}_i^I)^T \hat{\mathbf{\Sigma}}_I \mathbf{b}_i^R$$

$$= 2\delta_{i-k} \frac{\partial}{\partial \left[\mathbf{B}_R\right]_{jk}} (\mathbf{b}_k^I)^T \hat{\mathbf{\Sigma}}_I \mathbf{b}_k^R$$

$$= 2\delta_{i-k} (\mathbf{b}_k^I)^T \sigma_j^I$$

$$\left[\frac{\partial \mathbf{w}_3}{\partial \mathbf{B}_I}\right]_{ijk} = \frac{\partial \left[\mathbf{w}_3\right]_i}{\partial \left[\mathbf{B}_I\right]_{jk}} \overset{\text{(A.8)}}{=} 2\frac{\partial}{\partial \left[\mathbf{B}_I\right]_{jk}} \left[\text{diag}\left(\mathbf{B}_I^T \hat{\mathbf{\Sigma}}_I \mathbf{B}_R\right)\right]_i \tag{F.19}$$

$$= 2\frac{\partial}{\partial \left[\mathbf{B}_I\right]_{jk}} (\mathbf{b}_i^I)^T \hat{\mathbf{\Sigma}}_I \mathbf{b}_i^R$$

$$= 2\delta_{i-k} \frac{\partial}{\partial \left[\mathbf{B}_I\right]_{jk}} (\mathbf{b}_k^I)^T \hat{\mathbf{\Sigma}}_I \mathbf{b}_k^R$$

$$\overset{\hat{\mathbf{\Sigma}}_I = -\hat{\mathbf{\Sigma}}_I^T}{=} -2\delta_{i-k} \frac{\partial}{\partial \left[\mathbf{B}_I\right]_{jk}} (\mathbf{b}_k^R)^T \hat{\mathbf{\Sigma}}_I \mathbf{b}_k^I$$

$$= -2\delta_{i-k} (\mathbf{b}_k^R)^T \sigma_j^I$$

3. Combine $\left\{\frac{\partial \mathbf{w}_k}{\partial \mathbf{B}_{R/I}}\right\}_{k=1,2,3}$ with $\frac{\partial \mathcal{L}}{\partial \mathbf{w}}$ to obtain $\frac{\partial \mathcal{L}}{\partial \mathbf{B}} \in \mathbb{C}^{M \times N}$:

$$\left[\frac{\partial \mathbf{w}}{\partial \mathbf{w}_k}\right]_{ij} = \frac{\partial \left[\mathbf{w}\right]_i}{\partial \left[\mathbf{w}_k\right]_j} = \frac{\partial}{\partial \left[\mathbf{w}_k\right]_j} \left[\mathbf{w}_1 + \mathbf{w}_2 + \mathbf{w}_3\right]_i = \delta_{i-j} = \left[\mathbf{I}_N\right]_{ij}, \qquad k = 1, 2, 3 \tag{F.20}$$

$$\frac{\partial \mathcal{L}}{\partial \mathbf{w}_k} = \frac{\partial \mathcal{L}}{\partial \mathbf{w}} \frac{\partial \mathbf{w}}{\partial \mathbf{w}_k} \overset{\text{(F.12)}}{\underset{\text{(F.20)}}{=}} \sum_{l=1}^{L} \frac{\partial \mathcal{L}}{\partial \mathbf{s}^l}, \qquad k = 1, 2, 3 \tag{F.21}$$

$$\left[\frac{\partial\mathcal{L}}{\partial\mathbf{w}_1}\frac{\partial\mathbf{w}_1}{\partial\mathbf{B}_R}\right]_{jk} = \sum_{i=1}^{N}\left[\frac{\partial\mathcal{L}}{\partial\mathbf{w}_1}\right]_i\left[\frac{\partial\mathbf{w}_1}{\partial\mathbf{B}_R}\right]_{ijk} \overset{(\mathrm{F.16})}{=} 2\sum_{i=1}^{N}\left[\frac{\partial\mathcal{L}}{\partial\mathbf{w}_1}\right]_i \delta_{i-k}(\mathbf{b}_k^R)^T\sigma_j^R \quad \text{(F.22)}$$

$$= 2\left[\frac{\partial\mathcal{L}}{\partial\mathbf{w}_1}\right]_k (\mathbf{b}_k^R)^T\sigma_j^R = \left[2\hat{\boldsymbol{\Sigma}}_R^T\mathbf{B}_R\,\mathrm{diag}\left(\frac{\partial\mathcal{L}}{\partial\mathbf{w}_1}\right)\right]_{jk}$$

$$\overset{\hat{\boldsymbol{\Sigma}}_R=\hat{\boldsymbol{\Sigma}}_R^T}{=}\left[2\hat{\boldsymbol{\Sigma}}_R\mathbf{B}_R\,\mathrm{diag}\left(\frac{\partial\mathcal{L}}{\partial\mathbf{w}_1}\right)\right]_{jk}$$

$$\left[\frac{\partial\mathcal{L}}{\partial\mathbf{w}_2}\frac{\partial\mathbf{w}_2}{\partial\mathbf{B}_I}\right]_{jk} = \sum_{i=1}^{N}\left[\frac{\partial\mathcal{L}}{\partial\mathbf{w}_2}\right]_i\left[\frac{\partial\mathbf{w}_2}{\partial\mathbf{B}_I}\right]_{ijk} \overset{(\mathrm{F.17})}{=} 2\sum_{i=1}^{N}\left[\frac{\partial\mathcal{L}}{\partial\mathbf{w}_2}\right]_i \delta_{i-k}(\mathbf{b}_k^I)^T\sigma_j^R \quad \text{(F.23)}$$

$$= 2\left[\frac{\partial\mathcal{L}}{\partial\mathbf{w}_2}\right]_k (\mathbf{b}_k^I)^T\sigma_j^R = \left[2\hat{\boldsymbol{\Sigma}}_R^T\mathbf{B}_I\,\mathrm{diag}\left(\frac{\partial\mathcal{L}}{\partial\mathbf{w}_2}\right)\right]_{jk}$$

$$\overset{\hat{\boldsymbol{\Sigma}}_R=\hat{\boldsymbol{\Sigma}}_R^T}{=}\left[2\hat{\boldsymbol{\Sigma}}_R\mathbf{B}_I\,\mathrm{diag}\left(\frac{\partial\mathcal{L}}{\partial\mathbf{w}_2}\right)\right]_{jk}$$

$$\left[\frac{\partial\mathcal{L}}{\partial\mathbf{w}_3}\frac{\partial\mathbf{w}_3}{\partial\mathbf{B}_R}\right]_{jk} = \sum_{i=1}^{N}\left[\frac{\partial\mathcal{L}}{\partial\mathbf{w}_3}\right]_i\left[\frac{\partial\mathbf{w}_3}{\partial\mathbf{B}_R}\right]_{ijk} \overset{(\mathrm{F.18})}{=} 2\sum_{i=1}^{N}\left[\frac{\partial\mathcal{L}}{\partial\mathbf{w}_3}\right]_i \delta_{i-k}(\mathbf{b}_k^I)^T\sigma_j^I \quad \text{(F.24)}$$

$$= 2\left[\frac{\partial\mathcal{L}}{\partial\mathbf{w}_3}\right]_k (\mathbf{b}_k^I)^T\sigma_j^I = \left[2\hat{\boldsymbol{\Sigma}}_I^T\mathbf{B}_I\,\mathrm{diag}\left(\frac{\partial\mathcal{L}}{\partial\mathbf{w}_3}\right)\right]_{jk}$$

$$\overset{\hat{\boldsymbol{\Sigma}}_I=-\hat{\boldsymbol{\Sigma}}_I^T}{=}\left[-2\hat{\boldsymbol{\Sigma}}_I\mathbf{B}_I\,\mathrm{diag}\left(\frac{\partial\mathcal{L}}{\partial\mathbf{w}_3}\right)\right]_{jk}$$

$$\left[\frac{\partial\mathcal{L}}{\partial\mathbf{w}_3}\frac{\partial\mathbf{w}_3}{\partial\mathbf{B}_I}\right]_{jk} = \sum_{i=1}^{N}\left[\frac{\partial\mathcal{L}}{\partial\mathbf{w}_3}\right]_i\left[\frac{\partial\mathbf{w}_3}{\partial\mathbf{B}_I}\right]_{ijk} \overset{(\mathrm{F.19})}{=} -2\sum_{i=1}^{N}\left[\frac{\partial\mathcal{L}}{\partial\mathbf{w}_3}\right]_i \delta_{i-k}(\mathbf{b}_k^R)^T\sigma_j^I \quad \text{(F.25)}$$

$$= -2\left[\frac{\partial\mathcal{L}}{\partial\mathbf{w}_3}\right]_k (\mathbf{b}_k^R)^T\sigma_j^I = \left[-2\hat{\boldsymbol{\Sigma}}_I^T\mathbf{B}_R\,\mathrm{diag}\left(\frac{\partial\mathcal{L}}{\partial\mathbf{w}_3}\right)\right]_{jk}$$

$$\overset{\hat{\boldsymbol{\Sigma}}_I=-\hat{\boldsymbol{\Sigma}}_I^T}{=}\left[2\hat{\boldsymbol{\Sigma}}_I\mathbf{B}_R\,\mathrm{diag}\left(\frac{\partial\mathcal{L}}{\partial\mathbf{w}_3}\right)\right]_{jk}$$

$$\frac{\partial\mathcal{L}}{\partial\mathbf{B}_R} = \frac{\partial\mathcal{L}}{\partial\mathbf{w}_1}\frac{\partial\mathbf{w}_1}{\partial\mathbf{B}_R} + \frac{\partial\mathcal{L}}{\partial\mathbf{w}_3}\frac{\partial\mathbf{w}_3}{\partial\mathbf{B}_R} \quad \text{(F.26)}$$

$$\overset{(\mathrm{F.22})}{\underset{(\mathrm{F.24})}{=}} 2\left\{\hat{\boldsymbol{\Sigma}}_R\mathbf{B}_R\,\mathrm{diag}\left(\frac{\partial\mathcal{L}}{\partial\mathbf{w}_1}\right) - \hat{\boldsymbol{\Sigma}}_I\mathbf{B}_I\,\mathrm{diag}\left(\frac{\partial\mathcal{L}}{\partial\mathbf{w}_3}\right)\right\}$$

$$\overset{(\mathrm{F.21})}{=} 2\left\{\hat{\boldsymbol{\Sigma}}_R\mathbf{B}_R - \hat{\boldsymbol{\Sigma}}_I\mathbf{B}_I\right\}\mathrm{diag}\left(\sum_{l=1}^{L}\frac{\partial\mathcal{L}}{\partial\mathbf{s}^l}\right)$$

$$= 2\Re\left\{\hat{\boldsymbol{\Sigma}}\mathbf{B}\right\}\mathrm{diag}\left(\sum_{l=1}^{L}\frac{\partial\mathcal{L}}{\partial\mathbf{s}^l}\right)$$

$$\frac{\partial \mathcal{L}}{\partial \mathbf{B}_I} = \frac{\partial \mathcal{L}}{\partial \mathbf{w}_2}\frac{\partial \mathbf{w}_2}{\partial \mathbf{B}_I} + \frac{\partial \mathcal{L}}{\partial \mathbf{w}_3}\frac{\partial \mathbf{w}_3}{\partial \mathbf{B}_I} \tag{F.27}$$

$$\overset{\text{(F.23)}}{\underset{\text{(F.25)}}{=}} 2\left\{\hat{\boldsymbol{\Sigma}}_R \mathbf{B}_I \operatorname{diag}\left(\frac{\partial \mathcal{L}}{\partial \mathbf{w}_2}\right) + \hat{\boldsymbol{\Sigma}}_I \mathbf{B}_R \operatorname{diag}\left(\frac{\partial \mathcal{L}}{\partial \mathbf{w}_3}\right)\right\}$$

$$\overset{\text{(F.21)}}{=} 2\left\{\hat{\boldsymbol{\Sigma}}_R \mathbf{B}_I + \hat{\boldsymbol{\Sigma}}_I \mathbf{B}_R\right\} \operatorname{diag}\left(\sum_{l=1}^{L}\frac{\partial \mathcal{L}}{\partial \mathbf{s}^l}\right)$$

$$= 2\Im\left\{\hat{\boldsymbol{\Sigma}}\mathbf{B}\right\} \operatorname{diag}\left(\sum_{l=1}^{L}\frac{\partial \mathcal{L}}{\partial \mathbf{s}^l}\right)$$

$$\frac{\partial \mathcal{L}}{\partial \mathbf{B}} = \frac{\partial \mathcal{L}}{\partial \mathbf{B}_R} + j\frac{\partial \mathcal{L}}{\partial \mathbf{B}_I} = 2\hat{\boldsymbol{\Sigma}}\mathbf{B} \operatorname{diag}\left(\sum_{l=1}^{L}\frac{\partial \mathcal{L}}{\partial \mathbf{s}^l}\right) \tag{F.28}$$

### F.6 $\partial \mathcal{L}/\partial \boldsymbol{\tau}$

$$\left[\frac{\partial \mathbf{s}^l}{\partial \boldsymbol{\tau}}\right]_{ij} = \frac{\partial \left[\mathbf{s}^l\right]_i}{\partial \left[\boldsymbol{\tau}\right]_j} = \frac{\partial}{\partial \left[\boldsymbol{\tau}\right]_j}\left[\mathbf{u}^l + \mathbf{w} - \boldsymbol{\tau}\right]_i = -\delta_{i-j} = \left[-\mathbf{I}_N\right]_{ij}, \qquad l = 1,\dots,L \tag{F.29}$$

$$\frac{\partial \mathcal{L}}{\partial \boldsymbol{\tau}} = \sum_{l=1}^{L}\frac{\partial \mathcal{L}}{\partial \mathbf{s}^l}\frac{\partial \mathbf{s}^l}{\partial \boldsymbol{\tau}} \overset{\text{(F.29)}}{=} -\sum_{l=1}^{L}\frac{\partial \mathcal{L}}{\partial \mathbf{s}^l} \tag{F.30}$$

Combining eqs. (F.14), (F.28) and (F.30) leads to algorithms 1 and 2.

Figure G.1: Pyramic 48-element microphone array [16] used to acquire real-world dataset [10]. Eight microphones are mounted on six PCBs that form the edges of a tetrahedron.

# G  Real-data experiments (supplement)

Results in the main text present a summary of DeepWave's performance on two real-world datasets. The goal of this section is to provide a more elaborate description of the datasets, training process, and emphasise interesting observations.

## G.1  Dataset description

Two real-world datasets are considered:

**Dataset 1 [10]**  consists of a series of 92 microphone recordings from the *Pyramic*[16] array (fig. G.1) taken in an anechoic chamber to evaluate the performance of different direction-of-arrival algorithms [11]. Specifically, the dataset contains a series of 3 second recordings of human speech emitted by loudspeakers positioned around the edge of the chamber and located at the same height. Each recording has one, two, or three speakers active simultaneously. Recordings contain both male and female speech samples to cover a wide audible range.

**Dataset 2 [15]**  consists of a larger collection of microphone recordings from the *Pyramic*[16] array (fig. G.1) taken in an anechoic chamber. The goal of this dataset is to provide a generic dataset on which to evaluate the performance of array processing algorithms on real-life recordings with all the non-idealities involved. Specifically, the dataset contains 2700 recordings of human speech emitted from every direction of the anechoic chamber at a resolution of 2 degrees in azimuth and three different elevations ({-15, 0, 15} degrees). Recordings contain both male and female speech samples to cover a wide audible range. While the total number of recordings is significant, since each recording contains emissions from a single source, different audio samples can be combined to simulate complex multi-source sound fields. This data-augmentation task therefore allows us to assess the generalizability of DeepWave to such setups. Concretely, we construct a synthetic dataset of 5700 distinct microphone recordings with one, two, or three active speakers simultaneously.

## G.2  Data pre-processing

The raw time-series are pre-processed to get a suitable training set for DeepWave as follows:

- Instantaneous empirical covariances $\left\{ \hat{\Sigma}_t \right\}_t$ are obtained for 9 equi-spaced frequency bands spanning $[1500, 4500]$ Hz every 100 ms using *Short-Time Fourier Transforms (STFT)* [24, 8].

- APGD ground truths $\{\hat{\mathbf{x}}_t\}_t$ were estimated by solving eq. (3) with $\gamma = 0.5$, step size $\alpha = 1/\left\|\overline{\mathbf{A}} \circ \mathbf{A}\right\|_2^2$, and $\lambda_t = \max([\mathbf{x}_t^1]_1, \ldots, [\mathbf{x}_t^1]_N)/(\alpha\gamma)$, where $\mathbf{x}_t^1 \in \mathbb{R}^N$ is the APGD estimate obtained after one iteration of eq. (5).

After pre-processing, we obtain 2760 training samples $\mathcal{T} = \left\{\left(\hat{\boldsymbol{\Sigma}}_t, \hat{\mathbf{x}}_t\right)\right\}_t$ per frequency band for Dataset 1. The same process applied to Dataset 2 gives 151980 training samples $\mathcal{T} = \left\{\left(\hat{\boldsymbol{\Sigma}}_t, \hat{\mathbf{x}}_t\right)\right\}_t$ per frequency band.

### G.3 Network training

DeepWave is trained by solving eq. (9) using stochastic gradient descent (SGD) with momentum acceleration [22]. The optimisation problem is initialised as given in eq. (10). Dataset 1 is trained on an 80% random subset of $\mathcal{T}$ using mini-batches of size $N_{batch} = 100$, with the remaining 20% serving as a validation set. The learning rate was set to $10^{-8}$. Dataset 2 is also trained as above, except that 10 source directions are also witheld from the training set to assess how well DeepWave generalizes to emissions from unseen directions.

Regularisation parameters were chosen based on a grid search with optimal values $\lambda_{\boldsymbol{\theta}} = \lambda_{\mathbf{B}} = \lambda_{\boldsymbol{\tau}} = 0.1$. It was noticed during our experiments that regularising $\boldsymbol{\theta}$ and $\mathbf{B}$ provides little benefit to generalisation error and hence can be omitted. Regularisation of $\boldsymbol{\tau}$ is important however to ensure convergence to smooth biases. This is particularly relevant for rich acoustic fields where sources have weak spatial constraints, i.e. Dataset 2. (See also appendix H.)

Training and validation losses converged in less than 10 epochs for the optimal parameterisation, i.e. when $L = 5$ and $K$ ranges from 10 to 23 depending on the frequency band. Total training time for Dataset 1 was 10 minutes per band on an i7-8550U CPU with 32GB memory. Due to disk space constraints, Dataset 2 was trained on a dual-socket Intel E5-2680v3 with 256GB memory. Total training time for Dataset 2 was roughly 3 hours per band.

### G.4 Experimental results

In this section, we provide the supporting plots for the claims made in section 4 of the main paper:

- Figure G.2 shows DeepWave's learnt bias parameter on Dataset 1. Unlike APGD, the latter is highly nonuniform in space, and slightly stronger in magnitude.

- Figure G.3 shows the impulse response of DAS and DeepWave trained on Dataset 1 at 3.5 kHz, obtained by simulating the data from a single point-source in the field. Such plots were used to compute resolution scores of all algorithms across frequency bands.

- Figure G.4 shows example spherical fields obtained with DeepWave, DAS, APGD and APGD prematurely terminated applied to recordings in the validation set of Dataset 1. Resolution and contrast comparisons are moreover carried out. The true colour images displayed in fig. G.4 were obtained by mapping frequency channels into a colour spectrum (see the color-frequency mapping in fig. 3d).

- A video showing the evolution in time of DeepWave and DAS azimuthal sound fields (as in figs. 3a and 3b) is also available as supplementary material.[7]

# H  Further experiments in simulation

Results in the main text present a summary of DeepWave's performance on two real-world datasets. Though the datasets represent a particular real-world scenario (i.e. Dataset 1) and realistic complex sound fields (i.e. Dataset 2), the downside is that sound emissions are assumed to come from fixed spatial directions. It is therefore challenging to test DeepWave's generalisability on these datasets alone. The goal of this section is to investigate how well DeepWave generalises to richer datasets through simulation.

## H.1  Dataset description

The simulated dataset is designed to mimic a key application of acoustic cameras: accurate mapping of the sound field in an open-air setting from a given direction. To this end, the setup is modelled as follows:

- The scene is assumed to be a $120°$ spherical viewport in which sources are uniformly distributed.

- Source emissions follow a narrow-band point-source model at 2 kHz [23, 8], where their intensities are either uniform or Rayleigh-distributed with rate parameter $r = 1$. All images below show equi-amplitude visualisations only as they are easier to assess through visual inspection.

- Emissions from the scene are captured by a 64-element spherical microphone array of radius $r = 20$ cm.

- Empirical covariances matrices $\hat{\Sigma} \in \mathbb{C}^{64 \times 64}$ are synthesised using the traditional far-field measurement equation [23, eq.(12)] for point sources.

- APGD ground truths are obtained as described in appendix G.2.

Figure G.2: Bias parameter $\tau$ learnt by SGD run on Dataset 1 described in appendix G.1. We observe that the biasing is more prominent at sidelobes and around actual sources. This results in an increased angular resolution with fewer artefacts.

Figure G.3: Impulse response of DeepWave (top) vs DAS (bottom). We notice a shrinkage of the main lobe, resulting in increased angular resolution.

(a) DAS spherical sound field (resolution: 25.3° , RMS contrast: 0.78).

(b) DeepWave spherical sound field (resolution: 18.5° , RMS contrast: 0.97).

(c) APGD spherical sound field (resolution: 13° , RMS contrast: 0.97).

(d) APGD (terminated) spherical sound field (resolution: 21.4° , RMS contrast: 0.94).

Figure G.4: Intensity field reconstruction comparison between DAS, DeepWave ($L = 5$), APGD (converged, $N_{iter} = 17$), APGD (premature termination, $N_{iter} = 5$) on Dataset 1. In terms of resolution, DeepWave and APGD perform similarly, outperforming DAS by approximately 27%. The mean contrast scores for DeepWave and DAS over the test set of Dataset 1 are 0.99 ($\pm$0.0081) and 0.89 ($\pm$0.07), respectively. When limited to a number of iterations equal to the depth $L$ of DeepWave, APGD's performance degrades considerably.

The final dataset consists of 20'000 images that contain up to 10 sources in the field. Training the network is identical to appendix G.3, except for the batch-size which increases to 200 and the learning rate that is set to $10^{-7}$. In particular training converges in less than 10 epochs under an hour. The optimal parameterisation of the network is achieved with $L = 6$ and $K = 18$.

## H.2 Experimental results

- Figure H.1 shows example spherical fields obtained with DeepWave, DAS and APGD applied to recordings in the validation set of DeepWave.

- Figure H.2 investigates the influence of DeepWave's depth on the validation loss. Profiles show that 5 or 6 layers are sufficient for the investigated dataset.

- Figure H.3 investigates the runtime of APGD, DAS and DeepWave for different depths. DAS and DeepWave execute several orders of magnitude faster than APGD, regardless of network depths. Similar conclusions apply to Dataset 1 investigated in appendix G.

Figure H.1: Intensity field reconstruction comparison between (a) APGD ($N_{\text{iter}} = 48$), (b) DeepWave ($L = 5$), and (c) DAS. The image quality results corroborate with the observations made in fig. G.4.

Figure H.2: Influence of network depth on validation loss. The plot shows the relative squared-error on the validation set between APGD ground truth $\hat{\mathbf{x}}$ and DeepWave output $\mathbf{x}^L$ as a function of network depth $L$ using simulated data. The red curve corresponds to the full unconstrained dataset with up to 10 sources present in the field. The blue curve is obtained by retraining the network on a subset of the dataset where only up to 3 sources are present. Precision loss for small $L$ comes from insufficient sparsification of network output w.r.t. ground truth. On the other hand error increase for $L$ large are due to amplitude mismatches between ground truth and network output. This is presumably caused by the use of the rectified $\texttt{tanh}$ activation function to avoid gradient explosion during training.

| Method | $N_{iter}/L$ | Execution time [s] |
|---|---|---|
| APGD (converged) | 48 | $0.2118 \pm 24\text{e-}3$ |
| DeepWave | 6 | $0.0070 \pm 80\text{e-}6$ |
| DeepWave | 5 | $0.0065 \pm 95\text{e-}6$ |
| DeepWave | 3 | $0.0046 \pm 44\text{e-}6$ |
| DeepWave | 1 | $0.0031 \pm 46\text{e-}6$ |
| DAS | | $0.0020 \pm 41\text{e-}6$ |

Figure H.3: Runtime comparison of imaging methods on simulated dataset. Execution times averaged over 50 runs for a specific $\hat{\boldsymbol{\Sigma}} \in \mathbb{C}^{64 \times 64}$. DeepWave inference time is comparable to Delay-and-Sum and is adequate to obtain a fluid framerate on an acoustic camera. Runtimes in DeepWave weakly depends on network depth $L$ due to strong sparsity of the deblurring operator $\boldsymbol{\mathcal{D}}$: the main contributor to the former is evaluation of the backprojection term $\boldsymbol{\mathcal{B}}\,\text{vec}(\hat{\boldsymbol{\Sigma}})$. In stark contrast to DeepWave, APGD requires orders of magnitude more time to reach similar accuracy.

| $\theta$ | $\mathbf{B}$ | $\tau$ | $\mathcal{L}_{\text{test}}$ | rel. improv. [%] | rel. improv. [%] |
|---|---|---|---|---|---|
| ✗ | ✗ | ✗ | 0.160417 | | |
| ✗ | ✗ | ✓ | 0.054927 | 65.76 (✗✗✗) | |
| ✗ | ✓ | ✗ | 0.159698 | 0.45 (✗✗✗) | |
| ✓ | ✗ | ✗ | 0.159948 | 0.29 (✗✗✗) | |
| ✓ | ✗ | ✓ | 0.054910 | 65.77 (✗✗✗) | 0.03 (✗✗✓) |
| ✓ | ✓ | ✗ | 0.159234 | 0.74 (✗✗✗) | 0.29 (✗✓✗) |
| ✗ | ✓ | ✓ | 0.054917 | 65.77 (✗✗✗) | 0.02 (✗✗✓) |
| ✓ | ✓ | ✓ | 0.054900 | 65.78 (✗✗✗) | 0.03 (✗✓✓) |

Figure I.1: DeepWave performance comparison on simulated dataset described in appendix H.1 as a function of parameter degrees of freedom. A ✗ in the first three columns means that the associated parameter was frozen during training. In contrast a ✓ means that the parameter is optimized during training. $\mathcal{L}_{\text{test}}$ represents the data-fidelity loss term of eq. (9) evaluated over the test set. Finally, the last two columns show the relative improvement of $\mathcal{L}_{\text{test}}$ w.r.t. the baseline parameterisation given in parentheses. The results show that learning the shrinkage operator $\tau$ has the strongest net effect on improving predictive performance, while *for this setup* the deblurring $P_{\boldsymbol{\theta}}(\mathbf{L})$ and backprojection operators $\mathbf{B}$ provide marginal gains.

# I   Ablation study

Results in the main text and above present a summary of DeepWave's performance after optimal tuning of network parameters $\boldsymbol{\theta}$, $\mathbf{B}$, $\tau$ during training. Given the physical interpretation of these parameters as deblurring, backprojection and shrinkage operators respectively, we carry out an ablation study to investigate the relative importance of each parameter on DeepWave's ability to reconstruct ground truth APGD images.

Concretely, eight instances of DeepWave with $L = 6$ are trained on the simulated dataset described in appendix H.1. Each instance corresponds to a particular combination of free/frozen parameters such that all possible parameter triplets are taken into consideration. Frozen parameters remain at the initialisation point eq. (10) of SGD. Network performance is assessed by computing the data-fidelity term $\frac{1}{T} \sum_{t=1}^{T} \mathcal{L}_t$ from eq. (9) over the test set. The results are shown in Figure I.1.

As expected, freezing all three parameters (✗✗✗) produces the worst reconstructions as the network fails to converge to the ground truth after so few iterations. At the other end of the spectrum, learning all parameters (✓✓✓) leads to the best predictive performance, with a relative improvement of 65.78% over not learning anything. However the contributions of each parameter vary significantly: Learning $\tau$ (✗✗✓) has the strongest net effect (65.76%), whereas learning $\boldsymbol{\theta}$ (✓✗✗), $\mathbf{B}$ (✗✓✗) provide minimal gains over no learning (✗✗✗). The second half of Figure I.1 shows similar observations hold when training parameter pairs, where learning any parameter in addition to $\tau$ only provides small marginal gains over just learning the latter (✗✗✓). The reason for the marginal gains obtained when learning $\boldsymbol{\theta}$ and $\mathbf{B}$ is that the deblurring and backprojection operators are, for the specific experimental conditions investigated (point sources, non-reverberant environments (i.e. anechoic chambre), near-spherical geometries), very well modelled by initialisation scheme eq. (10). However, for environments containing reverberation and non-spherical array geometries, the observations above may differ significantly. In these contexts, learning $\boldsymbol{\theta}$ and $\mathbf{B}$ may lead to better predictive performance.

## Footnotes

[3]For example, two entries $(i, j)$ and $(k, l)$ in a 2D-array are neighbours if $\max(|i - k|, |j - l|) = 1$.

[4]At least in the context of sparse connectivity graphs explored here.

[5]which exists since the optimisation problem is convex.

[6]For notational simplicity, this section drops the subscript in $\mathcal{L}_t$.

[7]Also available online: `https://www.youtube.com/watch?v=PwB3CS2rHdI`