[Reviews · NeurIPS 2019]

Reviewer 1



The paper is well written and clearly organized. A smoother introduction of equation (2) would be nice for non-experts though. The quality of the theoretical part is undeniable. The elegant interpretations of the method steps given make the method quite compelling and the arguments convincing. As a simple comment, I must say that it's a shame that the PGD interpretation of the LISTA network is lost in practice when the ReLU activations are replaced by tanh. Moreover, the originality of the submission relies on the adaptation of a known technique (LISTA Network) for the particular application of real-time acoustic imaging. This can have great significance for practitioners and researchers in the field given the strong claims made in terms of performance and the disruptiveness of the ideas compared to the state-of-the-art approaches. For this reason, I believe that the experimental results section could have been larger, with more results to support the claims that are made by the authors. Indeed, I am surprised that 2760 data points were enough for training (without overfitting) and assessing the 5-layered neural network. More explanations on that matter would be nice. Also, I wonder what is the spherical map resolution N for the experiment and if the improvements stated in terms of runtime, resolution and contrast are averages over the 2760 * 0.2 = 552 test data points. In this case, some confidence measure of these scores could be useful to interpret the results. Other experiments from the supplementary material are cited in a small paragraph and should be further commented in the main document in my opinion. -- The authors have addressed most of my concerns. I am increasing my score under the condition that they add more precisions and results to their experimental section.

Reviewer 2



This work proposes a RNN for real-time reconstruction of acoustic camera spherical maps. It belongs to the wider topic of neural-networks inspired by optimization algorithms., e.g, LISTA based on iterative soft-thresholding algorithm. The authors did a good job in telling the stories behind the logic of the network they designed. They carefully reviewed the connection of the network with PGD, how in theory and by nature the network was developed and trained. The analysis is solid. However, I don't think the paper presents enough contribution and novelty. Firstly, the network was proposed to solve the acoustic imaging problem, and was not designed for a wider range of applications. Secondly, the formulation of the network was largely based on the natural settings of an acoustic imaging, e.g., the settings related to the spherical microphone arrays. Based on these settings, the authors discussed the theoretical justifications and their solutions, which seems not well applied to other areas.

Reviewer 3



+ Traditional acoustic camera methods had been advanced significantly with the advent of compressed sensing techniques, which reconstruct the original signals successfully by means of hand-crafted features or functions with nonlinear optimization, e.g., proximal gradient descent. However, the performance of the reconstruction process has been significantly slow due to nonlinear optimization steps. This paper proposes a new approach that substitutes the traditional nonlinear optimization approach with recurrent network architecture, i.e., by unrolling the iterative convex optimization algorithm in a form of neural network architecture. This paper takes a two-layered design, where a bias and back-projection gradient, and deblurring matrix are learned. + As described in the paper, the recurrent architecture has been proposed to substitute the signal reconstruction problem for other field applications of compressive sensings, such as compressive imaging. I'm not an expert in the acoustic camera field, but I cannot find out any existing works in this field yet. I, therefore, assume that this work is novel. + This submission is very carefully prepared. The supplemental document provides a thorough derivation of backpropagation and gradient descent details. The writing and soundness of this work should meet the standard of NeurIPS. + This work also tested the proposed method with real participants of eight people by recording their conversation. + I cannot find out any problem in terms of algorithms. - In this work, I cannot find out any ablation study of the parameters of the proposed method. It would be better if this paper might include an ablation study of the proposed method in terms of hyper-parameters or learnable parameters. - In addition, I cannot find out any quantitative comparison of the proposed method with state-of-the-art methods. For this reason, I cannot give a higher score to this work.

[Author Response · NeurIPS 2019]

We are grateful to the anonymous referees for their valuable comments and suggestions. We acknowledge that all referees correctly identified the main contributions of this work, namely: 1) a physically-inspired RNN generalising LISTA-like networks to acoustic imaging and related spherical array signal processing tasks; 2) a physically-inspired initialisation scheme for robust network training, supported by theoretical arguments; 3) a network architecture capable of handling complex-valued microphone outputs and with parameter size growing linearly with the pixel resolution. We address the comments of the reviewers below, denoted for brevity as **R1**, **R2**, **R3** respectively. Cited reference numbers correspond to those in the bibliography of the main paper.

**(R1.1)** *A smoother introduction of equation* (2) *would be nice.* **(R2.1)** *The paper might need to provide more background info on acoustic imaging.* Equation (2) is based on the well-known point-source data-model in far-field array signal processing, described in [52, Section 5.1] as well as [26]. The sparse imaging problem in [52, Section 5.6] is moreover very similar to (2). However, for completeness and as per the referees' advice, we shall include a brief derivation of (2) in the supplementary material of the final submission.

**(R1.2)** *PGD interpretation of the LISTA network is lost in practice when the ReLu activations are replaced by tanh.* To retain the PGD interpretation of the network, one can use truncated ReLus as activation functions. Given the chosen initialisation strategy, we will still converge with similar step sizes to those used with tanh non-linearities.

**(R1.3)** *I am surprised that 2760 data points were enough for training (without overfitting) and assessing the 5-layered network.* For this experiment the spherical maps have resolution $N = 2234$. As a result, the APGD images in the training set have total size $(2760 \times 0.8) \times 2234 = 4'932'672$. The total number of parameters in the 5-layered recurrent architecture is $23(K) + 2234(\dim(\tau)) + 2234 \times 48(\dim(\mathbf{B})) = 109'489$. There is hence 45 times more data than parameters to train, which is sufficient for the specific videoconference setup investigated to avoid overfitting. Indeed, as revealed by the analysis of algorithm 2, the training only affects parameters in $\mathbf{B}$ and $\tau$ which correspond to directions of nonzero intensity in the APGD ground truth images. Since source positions are in this case constrained to a few latitude/longitude pairs, this reduces further the total number of parameters to train. In more complex setups, where sources are unconstrained in location, the training set would of course need to be larger to avoid overfitting. Such a setup was investigated in appendix G where the training set was composed of 16'000 images.

**(R1.4)** *I believe that the experimental results section could have been larger, with more results to support the claims.* The ninth page will comment further on the experimental results. Furthermore, we will perform additional real-data experiments using a new comprehensive dataset, better suited for data-augmentation. See also answer **(R3.1)**.

**(R1.5)** *Some confidence measure of the [runtime, resolution, contrast] scores could be useful to interpret the results.* As per array signal processing standards, resolution was measured as the spread of the network impulse-response (see fig. F.3), obtained with synthetic data from a single source. The contrast and runtime scores were computed for the specific images shown in fig. 3 and fig. F.4. The final submission will provide average scores over the entire test set with confidence measures. For example, the average contrast scores in the videoconference setup are $0.99\,(\pm 0.0081)$ and $0.89\,(\pm 0.07)$ for DeepWave and DAS respectively (corresponding to a contrast improvement between 2 and 22%).

**(R2.2)** *The network was [tailored to] acoustic imaging, and not designed for a wider range of applications.* The focus on acoustic imaging stems from its importance in industrial applications, notably smart speaker assistants. However, equation (2) is based on a generic physical model common to many far-field array signal processing problems, including radar, sonar, interferometry, fault detection and medical imagery [26]. Since the submission, DeepWave has been succesfully tested in the context of radio astronomy with no architectural modifications. In addition, we stress that DeepWave is *not* restricted to spherical arrays. This assumption only motivates our initialisation scheme via proposition 1. As a matter of fact, the array used in fig. 3f is a tetrahedron.

**(R2.3)** *Is it possible to compare with the performance of LISTA?* We tried and this is not possible for the following reasons: 1) training proved impossible using the LISTA parametrisation (memory overflow even with sparsification as in [16]) and random initialisation; 2) without the Khatri-Rao parametrisation, the network output is not guaranteed to be real-valued given complex-valued raw microphone correlations.

**(R3.1)** *I cannot find out any ablation study of the parameters of the proposed method.* An ablation study has been conducted but left out of the manuscript due to space constraints. More specifically, we froze the training of network parameters, fixing them to their initial values provided in (10). All 6 combinations of free/frozen shrinkage, deblurring and backprojection operators were investigated. We also investigated the effect of the associated parameter regularisation terms in (9). We noticed that the shrinkage operator $\tau$ was most affected by regularisation and training. This is because the deblurring and backprojection operators are, for the specific experimental conditions investigated (point sources, anechoic chambre, near-spherical geometries), very well modelled by our initialisation scheme (10). For more complex environments and array geometries, the conclusions of the ablation study may however differ significantly. These results will be reported in the supplementary material of the final manuscript.

**(R3.2)** *I cannot find out any quantitative comparison of the proposed method with state-of-the-art methods.* Quantitative results in terms of runtime, resolution and contrast were provided for DeepWave and the state-of-the-art DAS algorithm for each experiment. We refer moreover the reviewer to **(R1.5)** regarding confidence measures on the scores.

**(R3.3)** *Stylistic comments.* We welcome these remarks and will take them into account for the final submission, while abiding with NeurIPS official stylistic guidelines.

[Meta-Review · NeurIPS 2019]

The paper proposes a RNN LISTA architecture to the problem of real-time acoustic imaging and a novel parametrization leading to a number of parameters that grow linearly wrt resolution instead of quadratically, as well as a novel initialization scheme. Some Experiments comparing this approach to the state-of-art in the field validate the proposed model. While the paper may of of narrow interest for the ML community, it presents some interesting contributions.